# Key epigenetic and signaling factors in the formation and maintenance of the blood-brain barrier

Jayanarayanan Sadanandan[1†], Sithara Thomas[1†], Iny Elizabeth Mathew[1], Zhen Huang[2], Spiros L Blackburn[1], Nitin Tandon[1], Hrishikesh Lokhande[3], Pierre D McCrea[4], Emery H Bresnick[5], Pramod K Dash[1], Devin W McBride[1], Arif Harmanci[6], Lalit K Ahirwar[1], Dania Jose[1], Ari C Dienel[1], Hussein A Zeineddine[1], Sungha Hong[1], Peeyush Kumar T[1*]

[1]The Vivian L. Smith Department of Neurosurgery, University of Texas Health Science Center McGovern Medical School, Houston, United States; [2]Departments of Neurology & Neuroscience, University of Wisconsin School of Medicine and Public Health, Madison, United States; [3]Department of Neurology, Harvard Medical School, Boston, United States; [4]Department of Genetics, TheUniversity of Texas MD Anderson Cancer Center, Houston, United States; [5]Wisconsin Blood Cancer Research Institute, University of Wisconsin School of Medicine and Public Health, Madison, United States; [6]UTHealth School of Biomedical Informatics, Houston, United States

*For correspondence:
peeyush.k.thankamanipandit@
uth.tmc.edu

[†]These authors contributed
equally to this work

Competing interest: The authors
declare that no competing
interests exist.

Reviewing Editor: K
VijayRaghavan, National
Centre for Biological Sciences,
Tata Institute of Fundamental
Research, India

## eLife Assessment

The specific questions taken up for study by the authors-in mice of HDAC and Polycomb function in the context of vascular endothelial cell (EC) gene expression relevant to the blood-brain barrier, (BBB)-are potentially **useful** in the context of vascular diversification in understanding and remedying situations where BBB function is compromised. The strength of the evidence presented is **incomplete**, and to elaborate, it is known that the culturing of endothelial cells can have a strong effect on gene expression.

**Abstract** The blood-brain barrier (BBB) controls the movement of molecules into and out of the central nervous system (CNS). Since a functional BBB forms by mouse embryonic day E15.5, we reasoned that gene cohorts expressed in CNS endothelial cells (EC) at E13.5 contribute to BBB formation. In contrast, adult gene signatures reflect BBB maintenance mechanisms. Supporting this hypothesis, transcriptomic analysis revealed distinct cohorts of EC genes involved in BBB formation and maintenance. Here, we demonstrate that epigenetic regulator's histone deacetylase 2 (HDAC2) and polycomb repressive complex 2 (PRC2) control EC gene expression for BBB development and prevent Wnt/β-catenin (Wnt) target genes from being expressed in adult CNS ECs. Low Wnt activity during development modifies BBB genes epigenetically for the formation of functional BBB. As a Class-I HDAC inhibitor induces adult CNS ECs to regain Wnt activity and BBB genetic signatures that support BBB formation, our results inform strategies to promote BBB repair.

## Introduction

Central nervous system vessels possess a BBB that prevents toxins and pathogens from entering the brain. A leaky BBB can lead to deleterious consequences for CNS disorders, including stroke,

traumatic brain injury, and brain tumors (*Profaci et al., 2020*). No treatment options are available to sustain and/or regenerate BBB integrity. Identifying and targeting the mechanism that forms and maintains the BBB is an attractive strategy to regain BBB integrity.

By comparing the EC transcriptome of peripheral and brain vessels, the genomic profile that contributes to BBB was described (*Daneman et al., 2010*; *Sabbagh et al., 2018*). Furthermore, extensive gene expression changes in CNS ECs during development were also reported (*Corada et al., 2019*; *Hupe et al., 2017*). However, molecular mechanisms governing BBB gene transcription and how such mechanisms contribute to BBB establishment and maintenance are unresolved.

Epigenetic modifications of histones and chromatin-modifying enzyme activities are critical determinants of gene expression (*Allis and Jenuwein, 2016*). HDACs associate with specific transcription factors and participate in gene repression (*Grunstein, 1997*). HDAC inhibitors are critical experimental tools for elucidating HDAC functions (*Li and Seto, 2016*), and four HDAC inhibitors are FDA-approved drugs for cancer treatment (*Yoon and Eom, 2016*). More than thirty HDAC inhibitors are being investigated in clinical trials (*He et al., 2022*). Similarly, PRC2, a complex of polycomb-group proteins (such as EZH2, EED, and SUZ12), represses transcription via its methyltransferase activity that catalyzes tri-methylation of H3K27 (*Cao et al., 2002*; *Kuzmichev et al., 2002*). An EZH2 inhibitor is FDA-approved for Follicular lymphoma (*Straining and Eighmy, 2022*).

Wnt/β-catenin (Wnt) signaling is essential for establishing the BBB (*Corada et al., 2019*; *Daneman et al., 2009*; *Hupe et al., 2017*; *Liebner et al., 2008*; *Stenman et al., 2008*), but the activity of this pathway declines gradually after that and is reported to be minimal in adults (*Liebner et al., 2008*; *Ma et al., 2013*; *Reis et al., 2012*). Evidence linking Wnt to regulating BBB genes includes an EC-specific knockout (KO) of β-catenin that affects BBB gene expression and integrity (*Gastfriend et al., 2021*; *Hupe et al., 2017*; *Tran et al., 2016*; *Wang et al., 2019*). In CNS ECs, Wnt signaling requires binding Wnt ligands Wnt7a/7b to Frizzled receptors. This interaction stabilizes the intracellular signaling molecule β-catenin by suppressing a cytoplasmic destruction complex, which would otherwise degrade β-catenin. Stabilized β-catenin translocates to the nucleus and regulates Wnt target gene transcription by interacting with DNA binding transcription factors TCF/LEF (T-cell factor/lymphoid enhancing factor). The mechanisms underlying the reduced Wnt signaling and how Wnt regulates the BBB gene transcription are not established.

We describe the discovery of epigenetic mechanisms responsible for regulating BBB gene expression during development, how Wnt signaling is reduced, and the significance of these mechanisms for BBB development. Furthermore, we demonstrated that HDAC inhibitors activate BBB gene cohorts expressed during BBB formation, suggesting a potential therapeutic intervention.

## Results

### Transcriptional downregulation of key BBB genes in adult cortical ECs: Evidence that distinct EC gene cohorts regulate BBB establishment vs maintenance

An intact, non-leaky BBB forms at E15.5 (*Ben-Zvi et al., 2014*). We hypothesized that the EC gene cohorts involved in BBB formation are expressed on E13.5, and 3–4- month-old adult CNS EC will express gene cohorts required for BBB maintenance. We defined transcriptomes of primary cortical ECs isolated from E13.5 and 3–4- month-old adults to test this (*Figure 1A*). mRNA-seq analysis revealed strong expression of EC genes, including *Pecam1*, *Cdh5*, and *Cldn5*, compared to perivascular cell types such as pericytes, neuronal and glial genes (*Figure 1—figure supplement 1A*). To define the gene cohorts responsible for BBB formation vs. BBB maintenance, we identified differentially expressed genes (DEGs) expressed in natural log (fold change) in E-13.5 relative to adult mice (*Figure 1A–B*). DEGs with a p-value <0.05 were considered significant. Compared to E13.5, 3738 genes were upregulated and 5650 genes were downregulated in adults (*Figure 1C*).

GO enrichment analysis of the DEGs identified five functional categories: angiogenesis, cell-to-cell junction, transporters, extracellular matrix, and DNA binding transcription factors (*Figure 1D*). Except for transporters, the other categories were characterized by more downregulated vs. upregulated genes. The DEGs included important BBB genes (*Figure 1E*). For example, tight junction (TJ) genes *Cldn1*, *Cldn5*, BBB transporters *Mfsd2a*, *Cav1*, and BBB-related transcription factors *Zic3*, *Foxf2*, and *Sox17* were differentially expressed (*Figure 1E*). The complete dataset is available on Geo under

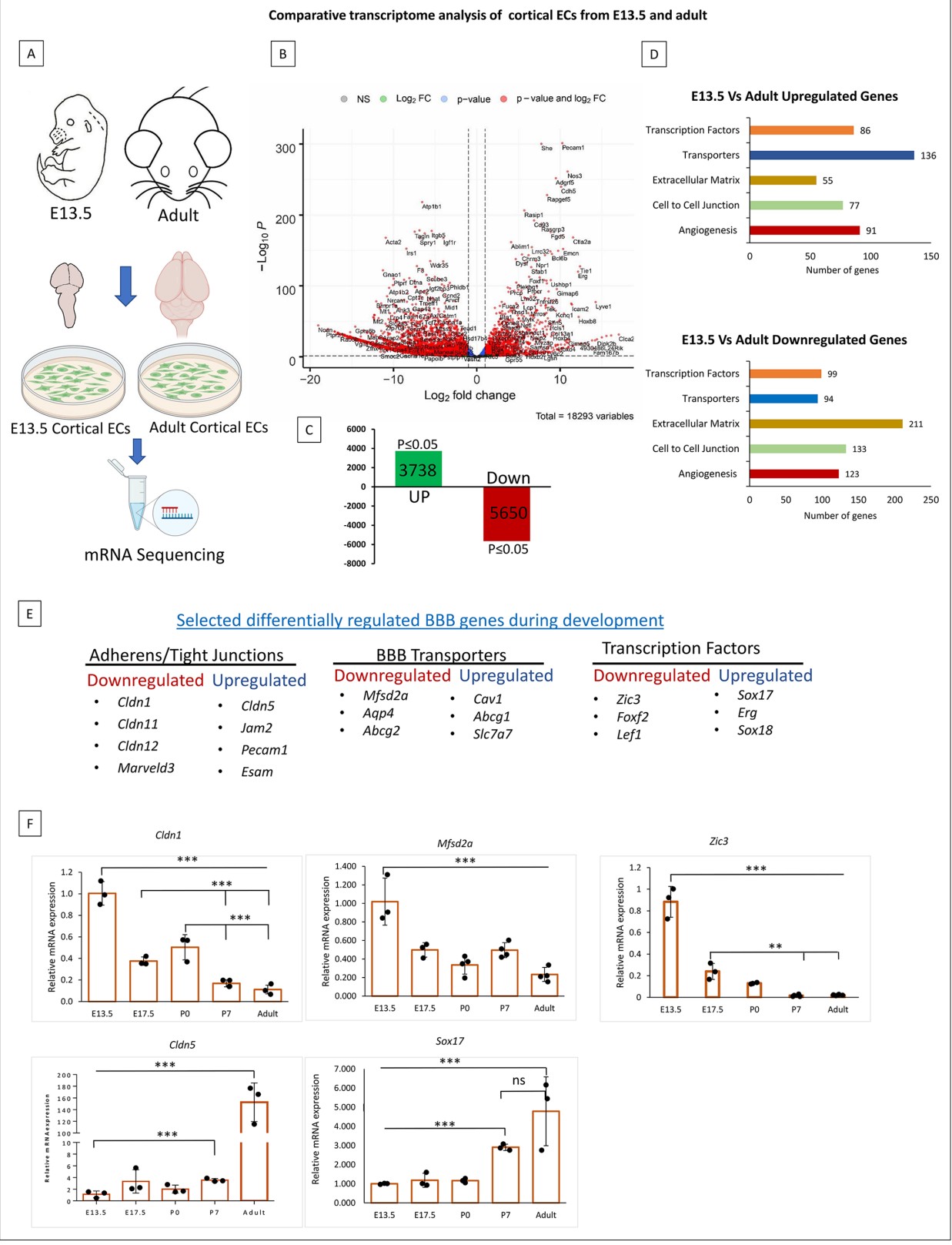

**Figure 1.** A distinct cohort of endothelial cells (EC) genes regulates the formation vs maintenance of blood-brain barrier (BBB). (**A**) Workflow for transcriptomic analysis. Primary ECs were isolated from E13.5 and adult (2–3-months-old) cortex, followed by RNA isolation, mRNA sequencing, and transcriptomic analysis. (**B**) Comparative transcriptome analysis of ECs from E13.5 and adult cortex. Volcano plot depicting downregulated and upregulated genes in adult primary cortical ECs compared to E-13.5. Genes marked in red are significant (p<0.05) N=3. (**C**) Diagram depicting the

*Figure 1 continued on next page*

*Figure 1 continued*

number of genes downregulated and upregulated in adult cortical primary ECs compared to E-13.5. (**D**) Downregulated and upregulated genes were categorized with five important EC functions. (**E**) Enrichment analysis revealed important BBB-related genes that were differentially regulated during development. (**F**) Relative mRNA expression of *Cldn1, Mfsd2a, Zic3, Sox17,* and *Cldn5.* in primary cortical ECs isolated from E13.5, E-17.5, P0, P7 and adult. Significant differences are observed between E-13.5 and all consecutive stages for *Cldn1*(***p<0.001, N=3/group), *Mfsd2a* (***p<0.001, N=3–4/group) and *Zic3*(***p<0.001 N=3–4/group). *Cldn1* showed significant differences between E-17.5 vs P7 and adult and P0 vs P7 and adult (***p<0.001 N=3/group). *Zic3* significantly differed between E-17.5 vs P7 and adult ECs (**p<0.01 N=3/group). Significant differences are observed between E-13.5 vs P7 and adults for *Cldn5* and *Sox17* (*p<0.001 N=3–4/group).

The online version of this article includes the following figure supplement(s) for figure 1:

**Figure supplement 1.** Endothelial cell markers expression and validation of mRNA seq data.

accession number GSE214923. Overall, the transcriptomic analysis identified EC gene cohorts that were expressed during the formation or maintenance of the BBB.

We validated the differential expression of important BBB and related genes (e.g. *Cldn1, Cldn5, Mfsd2a, Zic3,* and *Sox17*) at E-13.5, E-17.5, P0, P7, and in the adult. This analysis revealed that *Cldn1, Mfsd2a,* and *Zic3* expressions were significantly downregulated by E-17.5 and subsequent developmental stages (*Figure 1F*). Thus, the high expression of these genes might be required for the establishment and a baseline expression to maintain the BBB. By contrast, adult expression of *Cldn5* was significantly upregulated compared to other developmental stages, and *Sox17* was upregulated considerably compared to other developmental stages except for P7 (*Figure 1F*). Additionally, we validated the mRNA expression of *Cldn11* and *Foxf2* (*Figure 1—figure supplement 1B*).

*Hupe et al., 2017* demonstrated differentially regulated genes in brain ECs during embryonic development (*Hupe et al., 2017*). To further validate our data, we compared our transcriptome with the 1129 genes shown to be regulated during embryonic development. We found 315 of our downregulated transcripts in adults and 490 of our upregulated transcripts in adults matched with their dataset. Key overlapping genes are highlighted (*Figure 1—figure supplement 1C*).

## HDAC2 and PRC2 mediated transcriptional regulation of BBB genes

HDACs have critical roles in development and tissue homeostasis, and HDAC inhibitors are instructive experimental tools (*Delcuve et al., 2012*; *Marks, 2010*). To assess whether the transcriptional downregulation of BBB genes involves HDAC-dependent epigenetic repression, we treated adult ECs with a pan HDAC inhibitor trichostatin A (TSA) for 48 hr. TSA increased *Cldn1* (*Figure 2A*) and *Zic3* (*Figure 2—figure supplement 1*) expression relative to the control. Since there are four major HDAC classes, class I (HDAC 1, 2, 3, and 8), class II (HDAC 4, 5, 6, 7, 9, 10), class III (SIRTs 1–7), and class IV (HDAC 11), we tested whether specific HDACs mediate the repression. The class-I HDAC inhibitor MS-275 significantly upregulated *Cldn1, Mfsd2a,* and *Zic3* while downregulated *Cldn5* (*Figure 2A*, *Figure 2—figure supplement 1A*). No significant difference in *Cldn1* mRNA expression was observed with class-II HDAC inhibitor (*Figure 2—figure supplement 1A*). In adult ECs, HDAC2 exhibited greater expression than other class I HDACs (*Figure 2B*). To analyze the HDAC2 function, we utilized siRNAs to knockdown (KD) HDAC1, HDAC2, or HDAC3 in adult ECs. KD of HDAC2 significantly upregulated the repressed *Cldn1, Mfsd2a,* and *Zic3* genes and reduced *Cldn5* expression (*Figure 2C*, *Figure 2—figure supplement 1B*). By contrast, *Hdac1* and *Hdac3* KD did not affect these genes (not shown). BBB genes analyzed above were selected based on their expression patterns and to represent important functional attributes of BBBs, such as tight junctions (*Cldn1* and *Cldn5*), transporters (*Mfsd2a*), and transcription factors (*Zic3*). We used ChIP-qPCR to test whether HDAC2 directly regulates BBB gene expression in E13.5 and adult cortical EC contexts. Three primers were designed ((-) 500, TSS & (+) 500) spanning the 1 kb region on each side of the promoter. HDAC2 occupancy was detected in the (-) and (+) 500 regions of *Cldn1* in adults with no significant enrichment in E13.5. *Zic3* showed occupancy at all three regions (1 kb) in the adult stage, with E13.5 showing enrichment in TSS only (*Figure 2D*, *Figure 2—figure supplement 1B*). Conversely, HDAC2 occupied *Mfsd2a* only at E13.5((-) 500 & TSS), and in *Cldn5* occupancy was detected at both E13.5 (1kb) and in adults ((-) & (+) 500). These results link HDAC2 to the developmental control of BBB genes (*Figure 2D*, *Figure 2—figure supplement 1B*).

Our initial screening showed that the PRC2 inhibitor DZNEP significantly increased *Cldn1* expression in adult ECs (*Figure 2—figure supplement 1D*). To analyze the PRC2 function in this context,

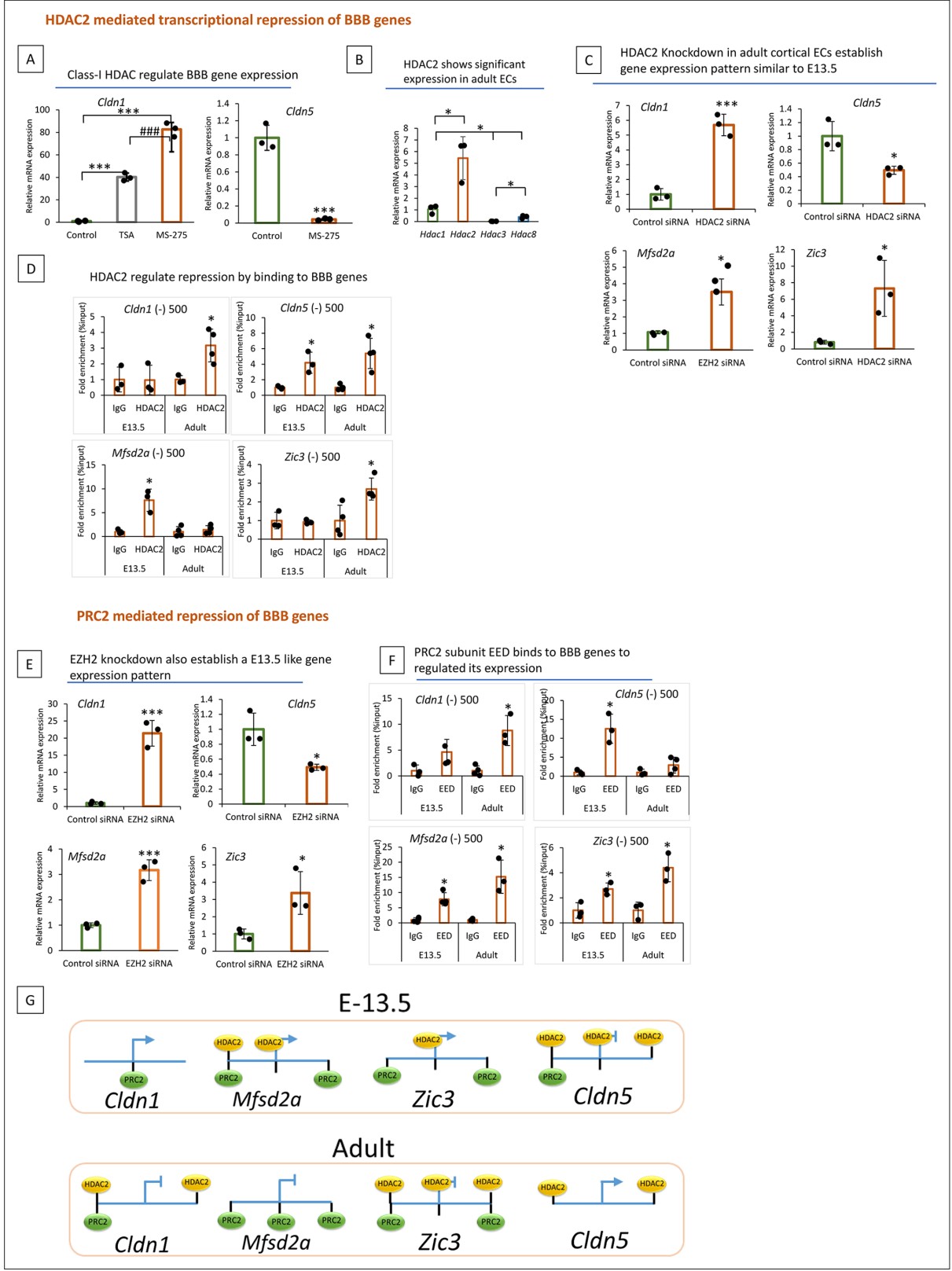

**Figure 2.** Epigenetic regulators histone deacetylase 2 (HDAC2) and polycomb repressive complex 2 (PRC2) regulate the transcription of blood-brain barrier (BBB) genes. (**A**) Quantitative PCR (qPCR) of adult primary cortical endothelial cells (ECs) treated with trichostatin A (TSA) (200 nm) and MS-275(10 um) 48 hr showed significantly increased mRNA expression of *Cldn1* compared to DMSO treated control while *Cldn5* was significantly decreased with MS-275 treatment when compared to control (*p<0.001 vs Control #p<0.001 vs TSA treatment N=3/group). (**B**) mRNA expression level of class-I

*Figure 2 continued on next page*

*Figure 2 continued*

HDAC family members in adult primary cortical ECs. Expression was normalized to housekeeping genes GAPDH and HDAC1. Significant mRNA expression of *Hdac2* was observed in adult primary cortical ECs compared to other Class-I HDACs (*p<0.05 vs HDAC1, 3, and 8 N=3/group). *Hdac1* showed significantly higher expression compared to HDAC3 and HDAC8 (*p<0.05) and HDAC8 showed significantly higher expression compared to HDAC3 (*p<0.05). (**C**) Effect of *Hdac2* siRNA on BBB gene expression in adult cortical ECs. Using lipofectamine adult cortical ECs were transfected with *Hdac2* siRNA (500 µg). qPCR analysis revealed that compared to control siRNA treated group *Hdac2* siRNA treated ECs showed significantly increased expression of *Cldn1*(*p<0.001), *Mfsd2a* (*p<0.05), and *Zic3* (*p<0.05) while *Cldn5* (*p<0.05) showed significantly decreased expression. N=3/group (**D**) HDAC2 occupancy of the indicated chromatin regions in primary cortical ECs from E-13.5 and adult. Occupancy was measured by ChIP followed by quantitative PCR (ChIP-qPCR). The adjacent gene and the distance to the TSS name chromatin regions. (* p<0.05 vs IgG N=3–4/group) (**E**) qPCR analysis of *Cldn1*, *Cldn5*, *Mfsd2a*, and *Zic3* in EZH2 and control siRNA-treated adult primary cortical ECs. Compared to the control siRNA-treated group, *Ezh2* siRNA-treated ECs showed significantly increased expression of *Cldn1* (*p<0.001), *Mfsd2a* (*p<0.001), and *Zic3* (*p<0.05), while *Cldn5* (* p<0.05) showed significantly decreased expression. N=3/group. (**F**) ChIP-qPCR analysis of PRC2 subunit EED on indicated chromatin regions and genes in E13.5 and adult primary cortical ECs. *p<0.05 vs IgG, N=3–4 / group. (**G**) Schematic representation of HDAC2 and PRC2 binding on indicated genes in cortical ECs at E-13.5 and adult.

The online version of this article includes the following source data and figure supplement(s) for figure 2:

**Figure supplement 1.** HDAC2 and PRC2 regulate the expression of key BBB genes during development.

**Figure supplement 1—source data 1.** PDF file containing original western blots for *Figure 2—figure supplement 1B, E*, indicating the relevant bands and treatments.

**Figure supplement 1—source data 2.** Original files for western blot analysis are displayed in *Figure 2—figure supplement 1B, E*.

we downregulated the PRC2 subunit EZH2 from adult ECs. Compared to the control, *Ezh2* siRNA KD significantly increased *Cldn1*, *Zic3*, and *Mfsd2a* expression and decreased *Cldn5* expression (*Figure 2E*, *Figure 2—figure supplement 1E*). ChiP-qPCR of PRC2 subunit EED revealed EED occupancy at various regions of *Cldn1*, *Mfsd2a*, and *Zic3* at E13.5 and in the adult (*Figure 2F*, *Figure 2—figure supplement 1F*). EED occupied *Cldn5* at E13.5, but not in adult ECs. Our data support a model in which HDAC2 and PRC2 are critical determinants of EC BBB transcriptomes during BBB development (*Figure 2G*).

## Distinct histone modifications delineate the transcription program of BBB genes

Since HDAC2 and PRC2 regulate multiple BBB genes, we sought to determine whether they share similar post-translational histone modifications. To examine this, we performed ChIP-qPCR of potentially involved histone marks, including the repressive marks H3K27me3, H3K9me3, and the activating marks H3K4me3 and H3K9ac. We have scanned approximately 1 kb genomic region surrounding the TSS of *Cldn1*, *Cldn5*, *Mfsd2a*, *Zic3*, and *Sox17* in E13.5 and adults. H3K9me3 ChIP-qPCR on our selected BBB genes didn't show significant binding in either stage (data not shown). We detected abundant enrichment of repressive histone mark H3K27me3 on *Cldn1*, *Mfsd2a*, and *Zic3* (*Figure 3A*, *Figure 3—figure supplement 1A-B*) in adult E.C.s compared to E13.5. *Cldn1*, *Mfsd2a*, and *Zic3* also showed significant enrichment of H3K27me3 compared to IgG in E13.5 (*Figure 3A*, *Figure 3—figure supplement 1A-B*). *Cldn5* showed significant enrichment of H3K27me3 in the 1 kb region of both E13.5 and adults (*Figure 3B*). However, E13.5 showed abundant enrichment of H3K27me3 in the −500 region compared to adults (*Figure 3B*).

Active histone mark, H3K4me3 showed significantly increased enrichment on *Cldn1* (TSS), *Mfsd2a* (1 KB), and *Zic3* (−500 and +500) at E13.5 compared to adult (*Figure 3C*, *Figure 3—figure supplement 1A-B*). *Cldn1* (±) 500 region didn't show any significant binding for H3K4me3 in both stages (*Figure 3C*) and the *Zic3* TSS region showed significant enrichment of H3K4me3 in the adult compared to E-13.5 (*Figure 3—figure supplement 1B*). Supporting its abundant expression in adults, *Cldn5* (1 KB) showed significant enrichment of H3K4me3 in adults compared to E13.5 (*Figure 3D*). Another active histone mark, H3K9ac, showed significant enrichment on *Cldn1* (TSS and +500) at E13.5 compared to the adult with no significant binding in the −500 region in both stages (*Figure 3E*). *Mfsd2a* showed significant enrichment of H3K9ac in the −500 and +500 regions at E-13.5 compared to adults. While the TSS region showed significant enrichment in adults compared to E-13.5 (*Figure 3—figure supplement 1A*). *Zic3* (TSS and +500) showed a significant binding for H3K9ac in E13.5 compared to the adult, while the −500 region didn't show any enrichment in both stages (*Figure 3—figure supplement 1B*). Distinct histone modifications in *Sox17* between E13.5 and adult CNS ECs

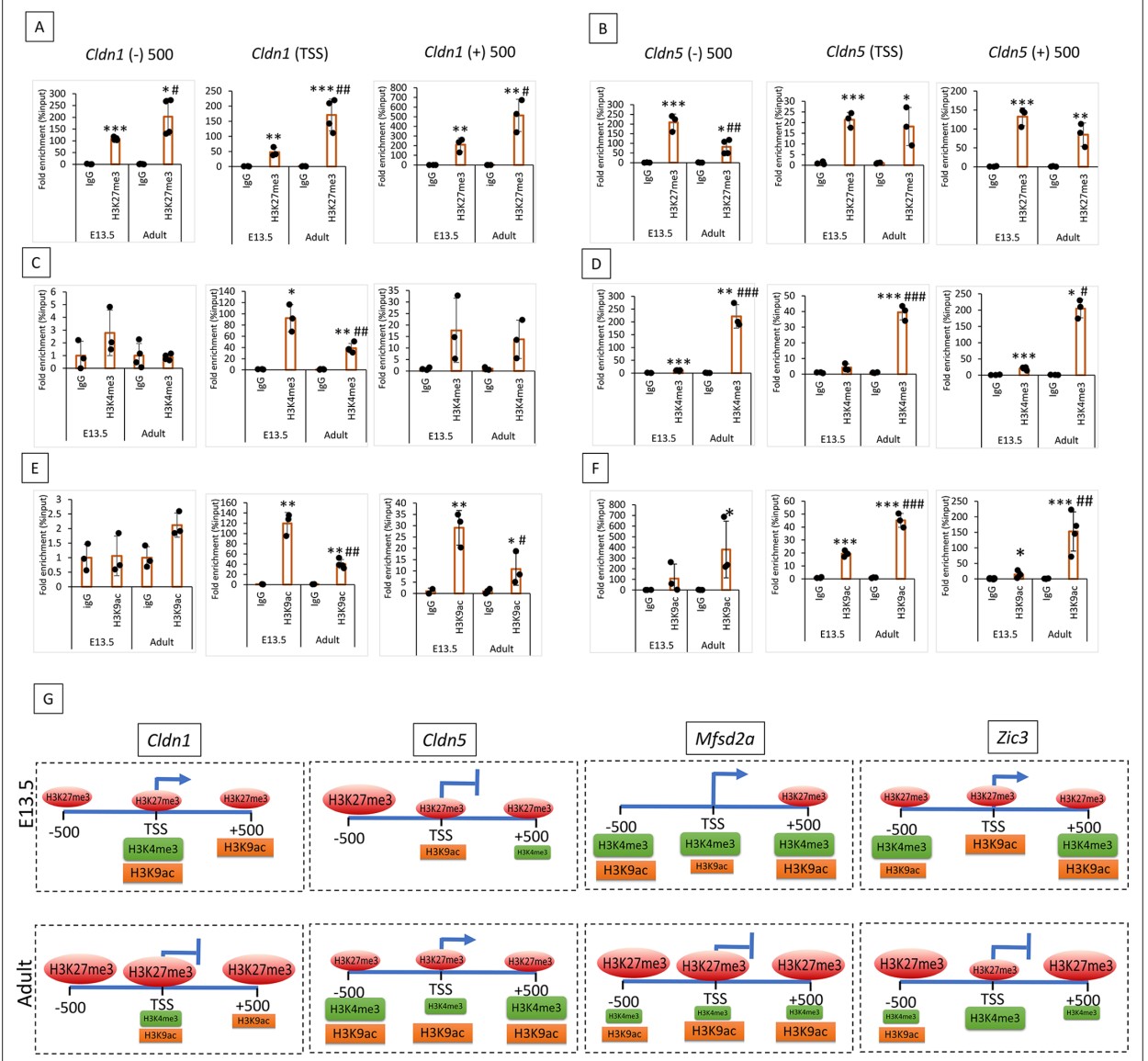

**Figure 3.** Blood-brain barrier (BBB) genes exhibit diverse post-translational histone modifications at E13.5 compared to adults. (**A, C & E**) ChIP followed by quantitative PCR (ChIP-qPCR) analysis of histone marks H3K27me3, H3K4me3, and H3K9ac on 1 KB region *Cldn1* gene in E13.5 and adult cortical endothelial cells (ECs). *Cldn1* gene was downregulated in ECs during development. The ChIP signals were normalized to IgG. (**B, D & F**) ChIP-qPCR showing the H3K27me3, H3K4me3, and H3K9ac density at the *Cldn5* gene in E13.5 and adult cortical ECs. *Cldn5* gene was upregulated during development. Data are shown as mean ± S.D. ***p<0.001, **p<0.01, *p<0.05 vs IgG & ###p<0.001, ##p<0.01, #p<0.05 vs respective E-13.5 histone mark. N=3–4/group. (**G**) Schematic representation of H3K27me3, H3K4me3, and H3K9ac binding density on *Cldn1*, *Cldn5*, *Mfsd2a*, and *Zic3* in E13.5 and adult primary cortical E.C.s. Shape size indicates the binding density in the indicated chromatin regions.

The online version of this article includes the following figure supplement(s) for figure 3:

**Figure supplement 1.** Key histone modifications in *Mfsd2a*, *Zic3*, and *Sox17*.

are shown in *Figure 3—figure supplement 1C*. A summary model of histone modifications in the 1 kb region of the BBB genes *Cldn1*, *Cldn5*, *Mfsd2a*, and *Zic3* is shown in *Figure 3G*. These results indicate that BBB genes acquire a unique epigenetic signature during development.

## HDAC2 activity is critical for the maturation of BBB, while PRC2 is dispensable

To examine the role of HDAC2 and PRC2 in BBB maturation, we KO *Hdac2* and PRC2 subunit *Ezh2* from ECs during embryonic development. To conditionally KO *Hdac2* or *Ezh2*, we used tamoxifen-inducible

*Cdh5(PAC)^CreERT2* mice. Tamoxifen was injected into the mother starting at E-12.5 and on alternate days until E-16.5 (*Figure 4A and E*). This allowed the KO of *Hdac2* or *Ezh2* before the maturation of the BBB.

It was observed that *Hdac2 and Ezh2* KO embryos grew normally and were alive on the day of sacrifice E17.5. Compared to WT, *Hdac2* ECKO pial vessels were dilated and showed increased angiogenesis (*Figure 4B*). BBB permeability was assessed in *Hdac2* ECKO mice using a 70KD FITC-conjugated dextran tracer. The tracer remained confined inside the vessels of E17.5 WT embryos, supporting previous findings that the BBB matures by E-15.5 (*Figure 4C*). Conversely, in *Hdac2* ECKO, dextran leaked into the cortical parenchyma (*Figure 4C*). Green fluorescent intensity measurements confirmed the immature BBB in the *Hdac2* ECKO (*Figure 4D*). A vascular analysis using angiotool determined that *Hdac2* ECKO had a significantly higher vascularized brain area than WT, indicating that HDAC2 also plays a key role in angiogenesis (*Figure 4D*). The pharmacological inhibition of class I HDAC using MS-275 in timed pregnant WT mice at E-13.5 showed significant tracer leakage into the brain parenchyma at E-15.5 compared to vehicle-treated control (*Figure 4—figure supplement 1A*). The MS-275 treated embryos also showed a thin cortex compared to the control, possibly due to the leakage of the drug into the brain which affects brain development (*Figure 4—figure supplement 1A*).

Compared to WT, at E-17.5 *Ezh2* ECKO showed dilated vessels with no evident increase in the pial vessel angiogenesis (*Figure 4F*). Confirming the loss of *Ezh2*, mRNA analysis on the whole brain showed a significant reduction. BBB permeability assay showed a subtle leakage of 70KD fluorescent tracer into the brain parenchyma (*Figure 4G*). However, the fluorescence intensity analysis showed no significant difference (*Figure 4H*). Furthermore, the mRNA analysis on the whole brain showed a significant decrease in *Ezh2* expression with no difference in *Cldn1*, *Mfsd2a*, and *Cldn5* (*Figure 4—figure supplement 1B*). Together, *Hdac2* and *Ezh2* ECKO data suggest that HDAC2 is critical for BBB maturation, whereas PRC2 is dispensable or a support mechanism required during later BBB development.

## Despite Wnt pathway activity in the adult, Wnt target genes are epigenetically repressed

Wnt pathway has been shown to influence BBB gene expression (*Hupe et al., 2017*). However, this pathway activity is reported to be minimal in adults (*Liebner et al., 2008*; *Ma et al., 2013*). In our transcriptomic analysis, 67% of Wnt signaling-related genes were downregulated in adults, while 33% were upregulated (*Figure 5A*). Interestingly, downstream Wnt target genes, including *Axin2*, *Lef1*, and *Vegfa* have downregulated in adult ECs, while the upstream Wnt pathway components like *Fzd4/6*, *Lrp5*, and *Ctnnb1* were upregulated (*Figure 5A*).

Since Wnt-regulated BBB genes such as *Cldn1*, and *Zic3* expression was also minimum in adults, we hypothesize that the Wnt pathway is still active in adult CNS ECs. In contrast, Wnt target genes are epigenetically repressed. To test this, we activated the Wnt pathway in E13.5 and adult ECs using identical concentrations of Wnt3a ligand or GSK3B inhibitor CHIR99021. mRNA analysis revealed that Wnt target genes *Axin2* and *Lef1* can be significantly activated in E13.5 ECs when treated with Wnt3a or CHIR99021, while no significant activation was observed in adults (*Figure 5B*). Further validating this finding, transcriptome analysis on adult ECs after Wnt3a treatment showed only activation of one Wnt target gene *Cd44* (*Figure 5C*). Intriguingly, the Wnt3a treatment downregulated 16 Wnt-related genes (*Figure 5C*). Next using immunohistochemistry of β-catenin in adult control and CHIR 99021 treated ECs we demonstrated that Wnt pathway activation could translocate or stabilize the Wnt transducer β-catenin into the nucleus (*Figure 5D*). These results indicate that the upstream Wnt pathway is active in adult CNS ECs.

Next, we investigated whether Wnt target genes are epigenetically repressed. Wnt target genes *Axin2* and *Lef1* were significantly upregulated in adult CNS ECs when treated with *Hdac2* or *Ezh2* siRNA, MS-275, and in *Hdac2* ECKO mutants (*Figure 5E*, *Figure 5—figure supplement 1A*). Furthermore, the *Axin2* promoter in adults showed significantly increased occupancy of HDAC2, repressive histone mark H3K27me3 in adults while active histone marks H3K4me3 and H3K9ac were reduced considerably compared to E-13.5 as analyzed by ChIP-qPCR (*Figure 5F*). *Lef1* also showed a similar repressive histone modification (*Figure 5—figure supplement 1B*). Furthermore, this was confirmed in vivo, as *Axin2* and *Lef1* mRNA expression were significantly increased in the E17.5 *Hdac2* ECKO cortex (*Figure 5—figure supplement 1B*). Additionally, treatment with MS-275 and LiCl (a Wnt

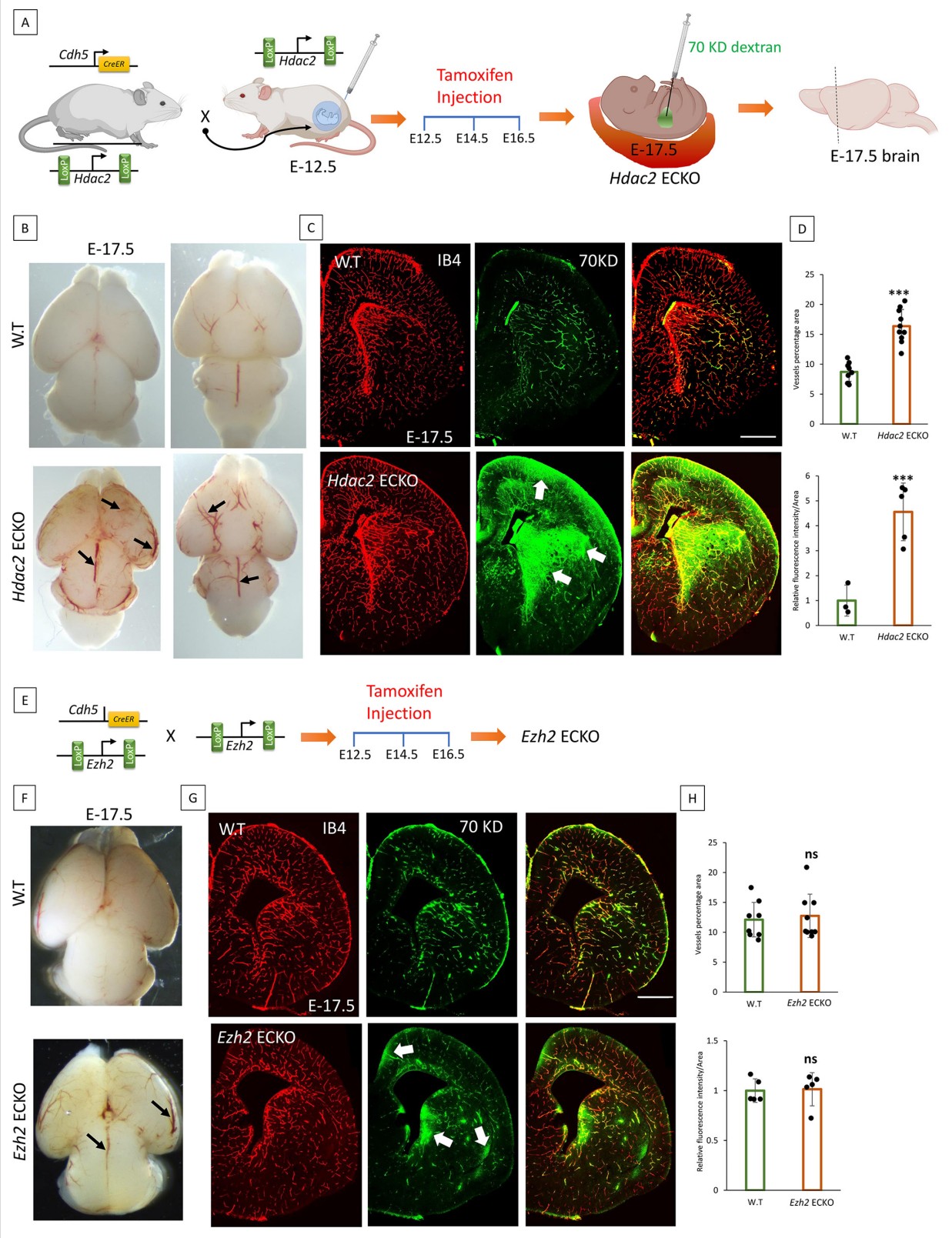

**Figure 4.** Histone deacetylase 2 (HDAC2) activity is required to form a functional blood-brain barrier (BBB), while polycomb repressive complex 2 (PRC2) is dispensable. (**A**) Effect of deletion of *HDAC2* from the endothelial cells (ECs) during embryonic development. Schematic representation of breeding scheme and generation of EC-specific KO of *HDAC2*. Tamoxifen was delivered to the pregnant mother at E-12.5, and brains were harvested at E-17.5. (**B**) Representative phase microscopy images of the dorsal surface and ventral surface of the brain at E-17.5. *Hdac2* ECKO shows a significant increase

*Figure 4 continued on next page*

Figure 4 continued

in pia vessels, as pointed out by black arrows on the dorsal surface. Black arrows in the ventral surface showed dilated vessels. (C) BBB permeability assay using 70KD tracer and isolectin B4 (IB4) staining to image the vessels. In *Hdac2* ECKO, a green fluorescent tracer leaked out of the vessels, as indicated by the white arrows. 10 x images are acquired and merged using tile scanning. Scale bar, 500 µm (D) Vessel percentage area of *Hdac2* ECKO was significantly higher than wild-type (WT) (*p<0.001 vs control N=10). A significant increase in fluorescent intensity was quantified in *Hdac2* ECKO compared to WT (*p<0.0001 vs control N=3 for WT and N=5 for *Hdac2* ECKO). (E) Schematic representation of breeding scheme and generation of *Ezh2* ECKO. Tamoxifen was injected the same as for *Hdac2* ECKO. (F) Representative phase microscopic image of WT and *Ezh2* ECKO. As black arrows show, the *Ezh2* ECKO brain shows dilated vessels with no visible increase in pial angiogenesis. (G) BBB permeability assay using 70KD-FITC Dextran shows subtle tracer leakage out of the vessels in *Ezh2* ECKO, represented by white arrows. IB4 staining reveals the vessels in the brain. 10 x images are acquired and merged using tile scanning. Scale bar, 500 µm (H) Quantification of vessel percentage area (WT N=9 *Ezh2* ECKO N=10) and fluorescent intensity didn't show any significant difference between WT and *Ezh2* ECKO (WT N=5 *Ezh2* ECKO N=5).

The online version of this article includes the following figure supplement(s) for figure 4:

**Figure supplement 1.** *Effect of class-I HDAC2 inhibitor on the embryonic mouse brain and the expression of Ezh2, Cldn1, and Cldn5 in Ezh2 ECKO mice.*

agonist) increased AXIN2 protein expression in adult cortical vessels, but this effect was not observed with LiCl treatment alone (*Figure 5—figure supplement 1C*). Thus, our data explain the mechanism behind the low Wnt pathway in adult CNS ECs.

## Low Wnt signaling epigenetically modifies the BBB genes to achieve BBB maturation

We investigated the relevance of the low Wnt pathway to BBB development. To this, we treated primary E13.5 cortical ECs with LF3 (inhibits the interaction of β-catenin and TCF4) for 48 hr. LF3 activity was confirmed by reduced mRNA expression of the Wnt target gene *Axin2* (*Figure 6A*). LF3 treatment induces adult/BBB maintenance-type gene expression patterns in E13.5 with a significant decrease in *Cldn1*, *Zic3*, and *Mfsd2a* expression and an increase in *Cldn5* and *Sox17* expression (*Figure 6A*). Another Wnt inhibitor IWR-1-endo also showed similar results (not shown), while β-catenin siRNA KD showed a similar gene expression pattern for *Axin2*, *Cldn1*, *Zic3*, and *Mfsd2a* and no difference in *Cldn5* (*Figure 6—figure supplement 1A*).

To determine whether Wnt pathway inhibition induces epigenetic modifications on its target genes and BBB genes, we performed Chip-qPCR analysis of HDAC2, EED, and histone mark H3K27me3, H3K4me3, and H3K9ac on the promoter of Wnt target genes *Axin2*, and *Lef1*, BBB genes *Cldn1*, *Cldn5*, *Mfsd2a,* and *Zic3*. An increased HDAC2 occupancy was observed on *Axin2*, *Lef1*, *Cldn1*, and *Mfsd2a* promoters when E13.5 ECs were treated with LF3 (*Figure 6B*, *Figure 6—figure supplement 1B*). *Zic3* and *Cldn5* showed no significant difference (not shown). EED and histone mark H3K27me3 showed an increased enrichment on the *Axin2 while Lef1* showed increased enrichment for H3K27me3 in the promoter after LF3 treatment (*Figure 6B*, *Figure 6—figure supplement 1B*) with no difference in other genes analyzed (not shown). *Cldn1* and *Mfsd2a* showed a significant decrease in the enrichment of active histone mark H3K4me3 after treatment with LF3, whereas *Cldn5* showed a significant increase (*Figure 6B*). Another active histone mark H3K9ac showed significantly increased enrichment on *Cldn5* (*Figure 6B*) with no difference in the promoter of other genes analyzed (not shown).

We then investigate the significance of the physiological reduction of Wnt signaling on BBB maturation. For this, we use *Ctnnb1^{lox/lox}*; *Cdh5^{CreERT2}* mice, widely used to attain the inducible EC-specific β-catenin gain of function (GOF) (*Figure 6C*). Upon tamoxifen treatment exon 3 of *Ctnnb1* (encoding β-catenin) will be deleted, leading to the expression of a stabilized form of β-catenin protein and thereby constitutive activation of canonical Wnt signaling (*Figure 6C*). Tamoxifen was injected into the pregnant mother, as illustrated in *Figure 6C*. At E-17.5, β-catenin GOF embryos showed significantly increased pial and periventricular angiogenesis (*Figure 6D–F*) compared to WT BBB permeability assay using 70KD FITC dextran revealed immature BBB in β-catenin GOF (*Figure 6E–F*). Further confirming the activation of the Wnt pathway in β-catenin GOF, *Axin2*, and *Cldn1* mRNA expression were significantly increased compared to the control (*Figure 6—figure supplement 1C*). Confirming this result, the pharmacological activation of Wnt signaling by the Wnt agonist LiCl also showed a significant BBB leakage compared to the control (*Figure 6—figure supplement 1D*). Our data suggest that a low Wnt pathway supports BBB maturations by epigenetically modifying BBB genes.

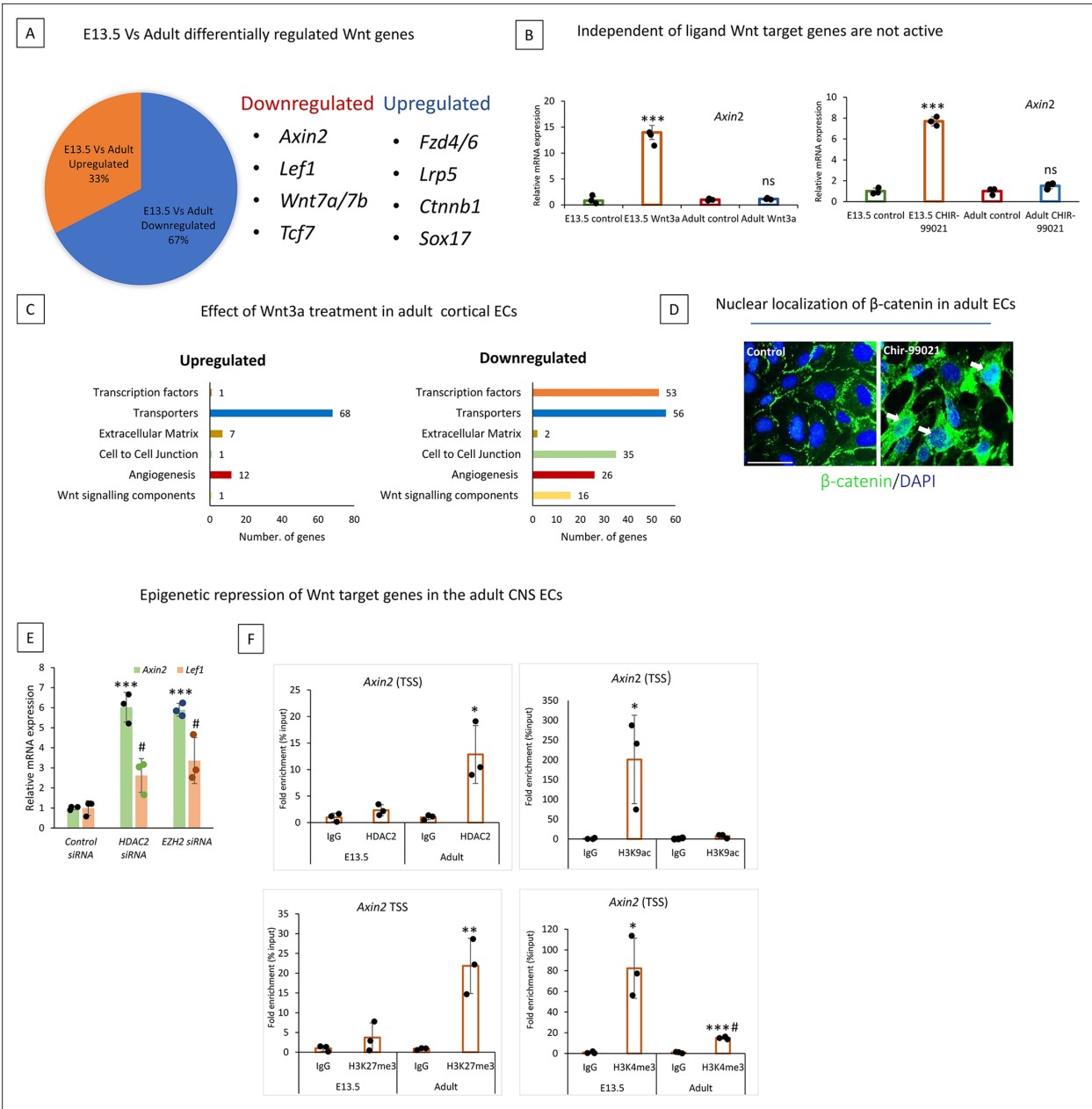

**Figure 5.** Upstream Wnt pathway is active in adult central nervous system (CNS) endothelial cells (ECs), but the Wnt target genes are epigenetically repressed. (**A**) Diagram depicting the percentage of Wnt-related genes downregulated and upregulated adult cortical primary ECs compared to E-13.5. Selected important Wnt-related genes are shown. (**B**) Ligand-independent transcriptional repression of Wnt target genes. In primary cortical ECs from E-13.5, activating the Wnt pathway with Wnt3a (200 ng/mL) or CHIR-99021 (5 uM) for 48 hr. caused increased mRNA expression of Wnt target genes *Axin2* and *Lef1* (measured via qRT-PCR). However, activation of the Wnt pathway in primary adult mouse brain ECs does not increase Wnt target gene expressions. *p<0.001 vs E-13.5 control, ns-no significant difference n=3/group. (**C**) mRNA sequencing was performed in control and Wnt3a (200 ng/mL) treated adult primary cortical ECs. Differentially expressed genes were categorized into six categories important to CNS endothelial cells. (**D**) Immunofluorescence staining of β-catenin (green) in control and Wnt agonist Chir-99021 treated endothelial cells. White arrows indicate the nuclear localization of β-catenin to the nucleus. 20 x images are acquired, cropped, and enlarged. Scale bar, 1 µm. (**E**) Adult primary cortical ECs transfected with control, *Hdac2* & *Ezh2* siRNA showed significant upregulation of Wnt target genes *Axin2.* ***p<0.001 vs control siRNA & #p<0.05 vs control siRNA. N=3/group (**F**) histone deacetylase 2 (HDAC2), histone marks H3K27me3, H3K4me3, and H3K9ac occupancy on the *Axin2* TSS regions in primary cortical ECs from E-13.5 and adult. ChIP followed by quantitative PCR (ChIP-qPCR) measured occupancy. N=3/group ***p<0.001, **p<0.01, *p<0.05 vs IgG & #p<0.05 vs H3K4me3 E-13.5.

The online version of this article includes the following figure supplement(s) for figure 5:

**Figure supplement 1.** Wnt target genes are epigenetically silenced in adult ECs.

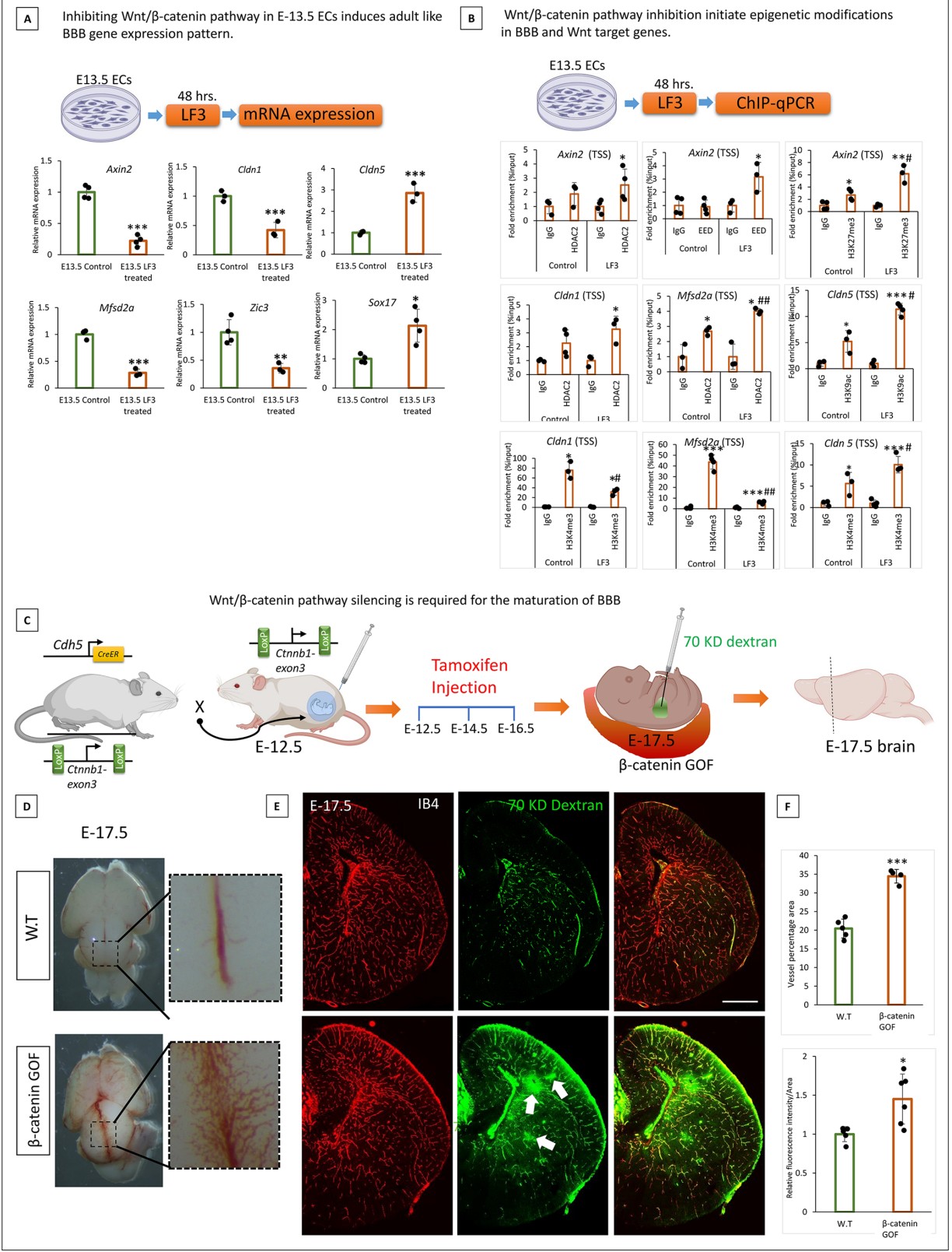

**Figure 6.** Low Wnt signaling epigenetically modifies the blood-brain barrier (BBB) genes to achieve BBB maturation. (**A**) Effect of Wnt pathway inhibition on BBB genes in E13.5 primary cortical endothelial cells (ECs). E-13.5 ECs were treated with LF3 (50 um) for 48 hr to inhibit the Wnt pathway. Significantly decreased mRNA expression of *Axin2* confirmed the reduced Wnt pathway. mRNA expression of *Cldn1*, *Mfsd2a*, and *Zic3* was significantly decreased *Cldn5* and *Sox17* expression was significantly increased after LF3 treatment. Data are shown as mean ± S.D. ***p<0.001, **p<0.01, *p<0.05 vs

*Figure 6 continued on next page*

*Figure 6 continued*

E13.5 control N=3/group. (**B**) Wnt pathway inhibition *via* LF3 induces epigenetic modifications in target gene *Axin2* and BBB genes *Cldn1*, *Mfsd2a*, and *Cldn5*. First row- *Axin2* showed significant enrichment of histone deacetylase 2 (HDAC2) and EED in LF3 treated E13.5 ECs compared to control (*p<0.05 vs IgG). Histone mark H3K27me3 showed significant enrichment in both conditions compared to IgG however, LF3-treated ECs showed significantly increased enrichment compared to the control. **p<0.01 vs IgG, #p<0.05 vs control N=3–4/group. Second row- LF3 treated E13.5 ECs showed significant enrichment of HDAC2 in *Cldn1* TSS (*p<0.05 vs IgG N=3–4/group). *Mfsd2a* showed significant enrichment in both conditions, while LF3 treatment showed a significantly increased enrichment compared to the control (*p<0.05 vs IgG & ##p<0.01 vs control N=3/group). *Cldn5* didn't show any significant difference in HDAC2 binding (not shown). At the same time, active histone marks H3K9ac showed significant enrichment in both conditions with an increased enrichment with LF3 treatment (***p<0.001, *p<0.05 vs IgG & #p<0.05 vs control N=3/group). Third row- H3K4me3 ChIP-qPCR on the TSS region of *Cldn1*, *Mfsd2a,* and *Cldn5* showed significant enrichment in both conditions with a decreased enrichment with LF3 treatment on *Cldn1* and *Mfsd2a* and an increased enrichment with LF3 treatment on *Cldn5*. ***p<0.001, *p<0.05 vs IgG & ##p<0.01, #p<0.05 vs control N=3–4/group. (**C**) Schematic representation of breeding scheme and generation of EC-specific gain of function (GOF) of β-catenin. Tamoxifen was delivered to the pregnant mother at E-12.5 and brains were harvested at E-17.5. (**D**) Representative Phase microscopy image of the dorsal brain from wild-type (WT) and β-catenin-GOF. Images in the square box were enlarged to show increased pial vessel angiogenesis. (**E**) BBB permeability assay using the 70KD FITC-Dextran tracer. Cortical vessels were stained using IB4.10X tile scanning images acquired and merged. FITC dextran was leaked out of the vessels in the brain of β-catenin GOF compared to WT 10 x images were acquired and merged using tile scanning. Scale bar, 500 µm (**F**) Quantification of brain vessels percentage area (***p<0.001 vs WT N=4–5/group) and green fluorescent intensity showed a significant increase in β-catenin GOF compared to WT ***p<0.001, *p<0.05 vs WT N=5–6/group.

The online version of this article includes the following figure supplement(s) for figure 6:

**Figure supplement 1.** Low Wnt signaling epigenetically modifies BBB genes to achieve a non-permeable BBB.

## Embryonic deletion of EC HDAC2 and Class-I HDAC inhibitor treatment of adult CNS ECs activate gene expression related to angiogenesis, BBB formation, and the Wnt pathway

Since the E-17.5 *Hdac2* ECKO showed a significant increase in vessel density and BBB leakage, we hypothesize this is due to the inefficient epigenetic repression of angiogenesis and BBB formation genes during development. To investigate this, we utilized E-17.5 *Hdac2* ECKO and FACS-sorted CD31 + ECs for ultra-low mRNA sequencing (*Figure 7A*). Indicating the critical role of HDAC2 in regulating the EC gene expression 2257 genes show significant upregulation and 1723 genes showed downregulation with a p-value <0.05.

We then performed functional classification of these genes based on key endothelial functions such as angiogenesis, cell-cell junctions, extracellular matrix, transporters, DNA-binding transcription factors, and the Wnt signaling pathway. Except for extracellular matrix function, all other categories were characterized by significantly upregulated genes (*Figure 7B*). Key differentially regulated genes are shown in *Figure 7C*. Critical angiogenesis genes, including *Vegfa* and *Eng*; tight junction proteins, including *Tjp1* (ZO-1), and *Ocln*; BBB transporters such as *Mfsd2a* and *Slc2a1* (GLUT1); and BBB-related transcription factors such as *Zic3*, *Foxf2*, and *Sox17* were upregulated (*Figure 7C*). Further supporting the HDAC2-mediated developmental regulation of the Wnt pathway, in *Hdac2* ECKO, several Wnt target genes, including *Axin2* and *Apcdd1*, were activated in E-17.5 ECs after *HDAC2* deletion (*Figure 7C*). Key genes that are downregulated include *Apoe*, *Psen1,* and *Vegfb*. We didn't observe any difference in the expression of *Cldn5*. However, *Cldn1* shows significantly lower reads, warranting deeper sequencing.

To identify the potential of inhibiting HDAC2 in activating BBB and angiogenesis-related genes in adult ECs, and to obtain a detailed picture of CNS EC genes regulated by MS-275, a class-I HDAC inhibitor, we performed mRNA-seq analysis of MS-275-treated primary adult cortical ECs in culture and compared with the control. Compared to the control, MS-275 treatment upregulated 3436 genes and downregulated 3050 genes in adult cortical ECs (*Figure 7D*). Differentially expressed genes were grouped into six relevant categories, and all the categories showed a significant no.of differentially regulated genes (*Figure 7E*). *Figure 7F* illustrates the potential of MS-275 in regaining the expression of BBB, angiogenesis, Wnt, and transcription factors that are differentially regulated during development (E-13.5 vs Adult).

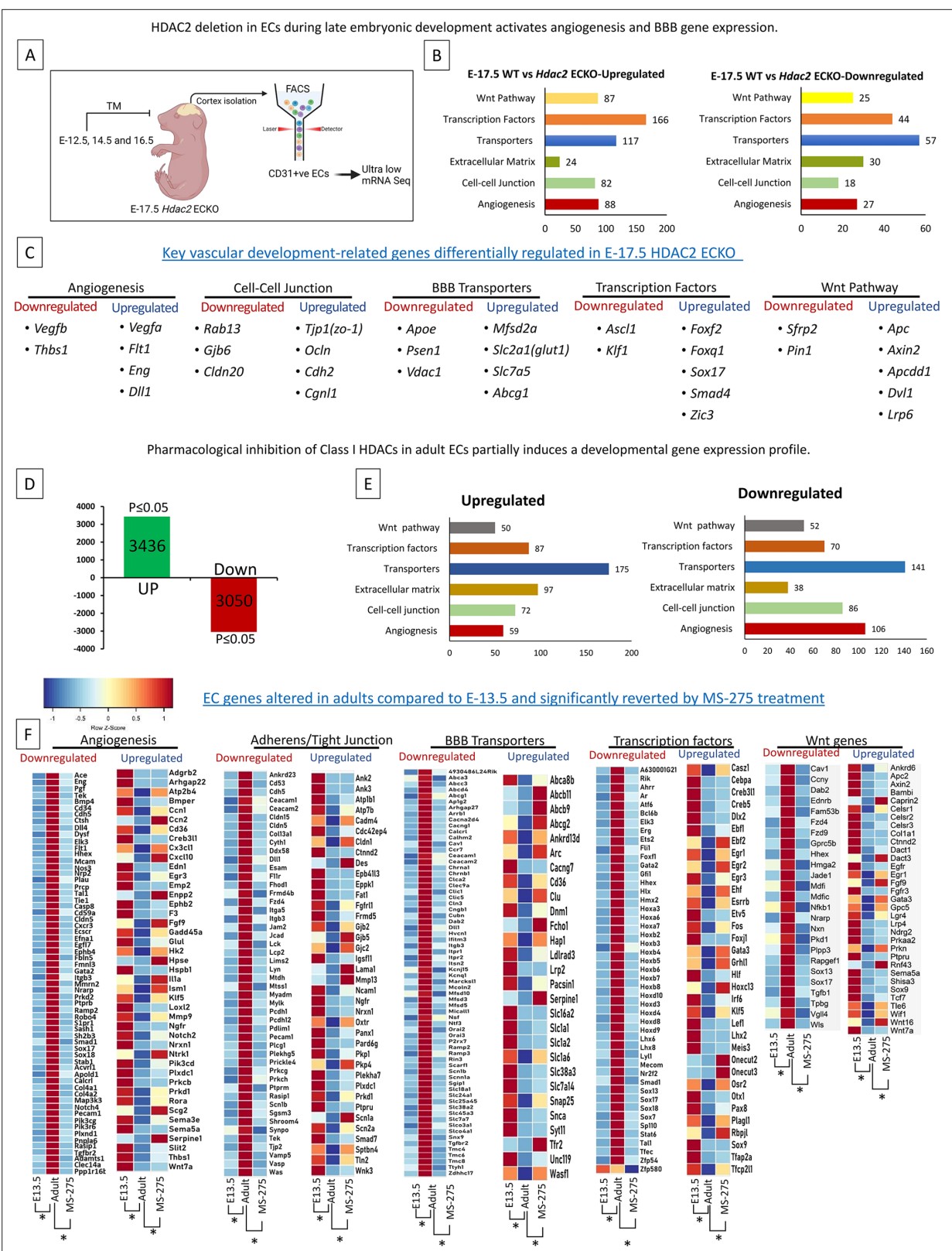

**Figure 7.** Histone deacetylase 2 (HDAC2) deletion or inhibition activates the angiogenesis, blood-brain barrier (BBB), and Wnt pathway genes.
(**A**) Tamoxifen was administered to pregnant mothers starting at E-12.5 on alternate days until E-16.5. The cortex was harvested at E-17.5, and CD31⁺ endothelial cells were isolated via fluorescence-activated cell sorting (FACS). The resulting cells were processed for ultra-low-input mRNA sequencing.
(**B**) Six key EC-regulated pathways categorize downregulated and Upregulated genes. (**C**) Key differentially expressed genes in EC-regulated pathways.

*Figure 7 continued on next page*

*Figure 7 continued*

N=3 from three different mothers (*p<0.05). (**D**) In adult ECs, treatment with MS-275 induces partial reactivation of angiogenesis and BBB formation supporting gene cohorts. Diagram depicting the number of genes downregulated and upregulated in MS-275 treated adult cortical ECs compared to control. (**E**) Downregulated and upregulated genes were categorized with six important EC functions. (**F**) Heat map of expression values (Z score) for differentially expressed genes (*p adj <0.05) in E13.5, adult Control, and adult treated with MS-275. Five gene categories showing significant differences between E13.5 vs adult control and Adult Control vs Adult MS-275 treatment are presented. N=3 for E13.5 and adult control, N=4 for MS-275.

The online version of this article includes the following figure supplement(s) for figure 7:

**Figure supplement 1.** *Wnt7a* expression is suppressed in adult CNS ECs, but epigenetic activation is possible.

## Pharmacologically induced epigenetic changes are reversible and partially translate to human vessels

We next assessed whether the gene expression changes acquired by adult ECs following MS-275 treatment were reversible. To this end, we treated adult ECs with MS-275 for 48 hr and collected ECs at 48 hr of treatment and 7 d after treatment. As previously shown, *Cldn1* was significantly upregulated,

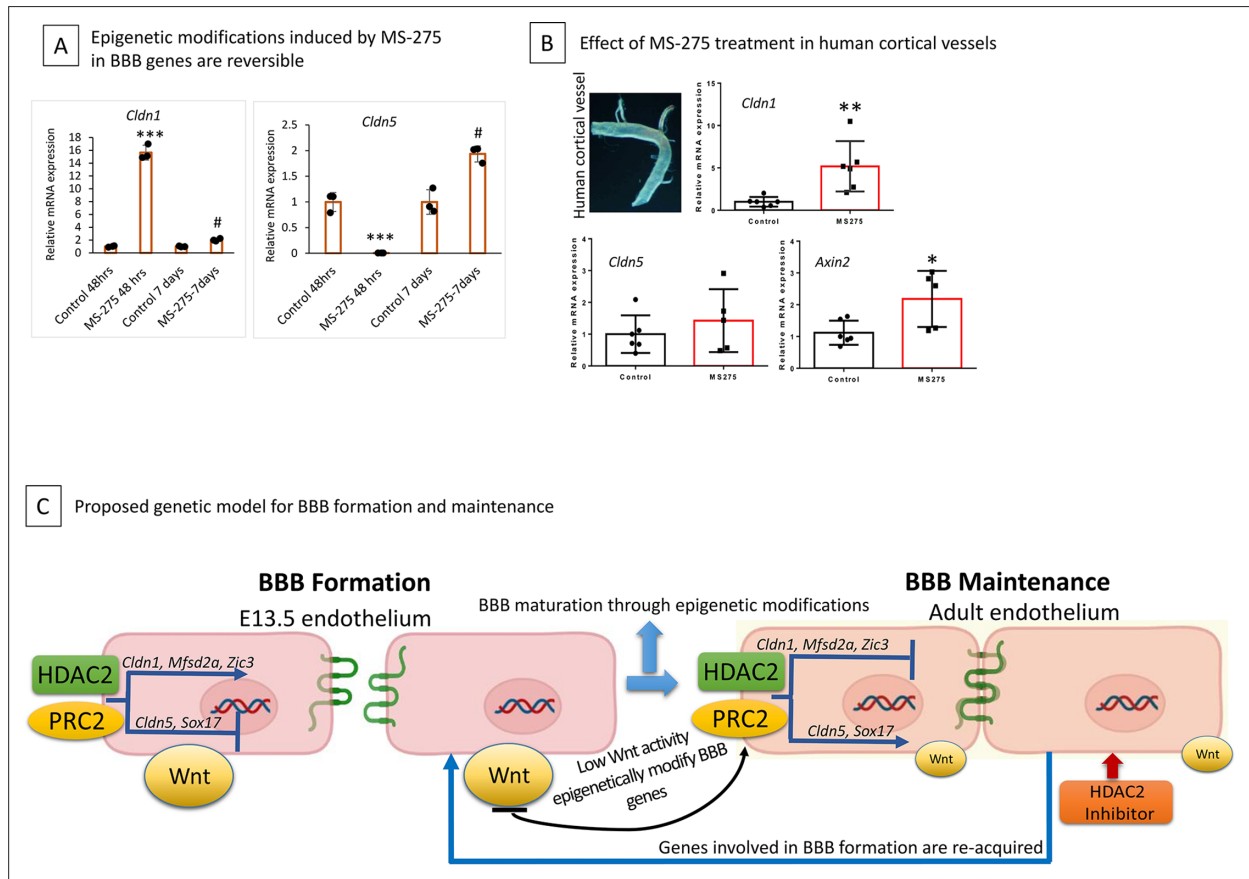

**Figure 8.** Epigenetic modification induced by MS-275 is temporary. (**A**) Adult primary cortical endothelial cells (ECs) treated with MS-275 for 48 hr showed significant activation of *Cldn1* and downregulation of *Cldn5*. mRNA analysis on ECs after 7 d of withdrawing MS-275 showed a significant reversal of expression back to normal. *p<0.001 compared to control 48 hrs. # p<0.001 compared to MS-275 48 hr (**B**) Effect of class-I histone deacetylase (HDAC) inhibition on adult human cerebral arteries. Representative phase contrast image of human temporal lobe vessels in culture collected from epilepsy patients undergoing surgery. MS-275 treated human vessels showed significantly increased mRNA expression of *Cldn1* and *Axin2* compared to control. **p<0.01 and *p<0.05 N=5/group. No significant difference was observed in *Cldn5* expression. (**C**) Schematic diagram illustrating the mechanisms underlying blood-brain barrier (BBB) formation and maintenance led by epigenetic regulators histone deacetylase 2 (HDAC2) and polycomb repressive complex 2 (PRC2). HDAC2 and PRC2 epigenetically repress EC gene cohorts that support BBB formation during development. Active Wnt signaling supports the expression of gene cohorts required for BBB formation. In contrast, a reduction in Wnt signaling recruits HDAC2 to these gene cohorts to support the formation of a functional or intact BBB. Inhibiting HDAC2 in adult ECs induces the reacquisition of gene cohorts that support BBB formation, thus representing a potential therapeutic opportunity to repair a damaged BBB.

and *Cldn5* was downregulated considerably after 48 hr (*Figure 8A*). While the expression of *Cldn1* and *Cldn5* in adult ECs returned to normal in ECs collected 7 d after the treatment (*Figure 8A*).

Finally, we examined if human vessels show similar reactivation when treated with MS-275. To this end, the human brain vessels collected from epilepsy surgery irrespective of age and gender. We tested three genes with MS-275 treatment: *Cldn1*, *Cldn5*, and *Axin2*. We found that MS-275 treatment significantly activates the expression of *Cldn1* and *Axin2* compared to the control, with no difference in *Cldn5* expression (*Figure 8B*).

The graphical summary of the proposed model for epigenetically controlled BBB formation and maintenance is illustrated in *Figure 8C*.

## Discussion

The mechanisms that create and maintain the BBB are vitally important but poorly understood. We identified EC gene cohorts that differentially support BBB formation and maintenance, described the genetic and signaling mechanisms that establish the BBB, and presented an attractive strategy to activate gene cohorts involved in BBB formation that may promote BBB repair.

*Mfsd2a* is required for BBB formation (*Ben-Zvi et al., 2014*), and the transcription factors *Zic3* and *Foxf2* can induce BBB markers even in peripheral ECs (*Hupe et al., 2017*). Our transcriptomic analysis revealed that *Cldn1*, *Mfsd2a*, *Zic3*, and *Foxf2* were significantly expressed in E13.5 compared to adults, suggesting their requirement during BBB formation. Increased levels of *Cldn5*, *Pecam*, *Abcg1*, and *Sox17* in adult CNS ECs point to its significance in BBB maintenance. The results of our study partially agree with those of prior gene expression studies (S.*Figure 1C*), but provide new concepts regarding downstream epigenetic mechanisms governing BBB gene expression. Although a high purity level is achieved in primary EC culture, we cannot exclude the possibility that transcripts from other cell populations may be identified. Even though we used similar culture conditions for both embryonic and adult cortical ECs, culture-induced changes have been reported previously (*Sabbagh and Nathans, 2020*) and should be considered as a varying factor when interpreting our results.

It is not known how BBB gene transcription is regulated. In CNS ECs, HDAC2 and PRC2 directly regulate the transcription of important BBB genes, including *Cldn1*, *Cldn5*, *Mfsd2a*, and *Zic3*. While HDAC2 and PRC2 commonly occupy repressed genes, we detected them in both active and repressive states at BBB genes. This result is consistent with the established dual transcriptional role of these epigenetic regulators (*Ferrari et al., 2014*; *Greer et al., 2015*; *Jahan et al., 2018*; *Kaneko et al., 2013*; *Somanath et al., 2017*; *Wang et al., 2017*; *Wang et al., 2009*; *Zupkovitz et al., 2006*). Furthermore, we present evidence that HDAC2 is required to form a functional BBB and induce anti-angiogenic signals. Even though the loss of vascular integrity and lethality at E-13.5 was reported in non-inducible conditional *Ezh2* KO using *Tie2* Cre-mouse (*Delgado-Olguín et al., 2014*), loss of *Ezh2* from E-13.5 did not significantly affect BBB permeability. These results indicate that, during the differentiation phase, HDAC2 initiates the transcriptional control, and PRC2 functions in a supporting mechanism. However, the KD of these regulators from adult CNS ECs induces a similar gene expression pattern, indicating the possibility that PRC2 facilitates the epigenetic modification during later BBB development and maintenance.

Since we did not detect H3K9me3, the repression of genes involved in BBB formation is mainly mediated through H3K27me3. On varying abundance, repressive histone mark H3K27me3 and active histone mark H3K4me3 were detected in our selected BBB genes, suggesting that repressive and activating histone methylation marks modify the promoter. Bivalent histone states can correlate with genes transcribed at low levels, suggesting these genes are poised for activation. In addition, we detected H3K9ac at the active and repressed BBB genes. Bivalent promoters can harbor H3K9ac. Thus, among the five BBB genes analyzed, each gene exhibits a different epigenetic signature, defined by histone acetylation and methylation. These results systematically demonstrate the complexity of epigenetic regulation in BBB genes and the evidence of unique epigenetic signatures. The data presented here does not entirely represent all the histone modifications on BBB genes, as other essential histone modifications, such as H3K27ac and H3K14ac, have not been examined.

The mechanism that confers low Wnt signaling in adult CNS ECs is unknown. Our results demonstrate that Wnt pathway components are still active in adult CNS ECs, yet the Wnt target genes are epigenetically inactive. Thus, in the adult CNS ECs, Wnt pathway activation permits stabilized β-catenin to enter the nucleus. Whether this nuclear β-catenin binds to Wnt target genes is unknown.

Nevertheless, we demonstrated that HDAC2 and PRC2 repress Wnt target genes. Other studies have revealed that the basal or minimal Wnt pathway maintains adult BBB integrity, but its activation also prevents stroke-induced BBB damage. This also suggests the possibility of switching Wnt-regulated genes during development. Our transcriptomic analysis revealed the Wnt-regulated adult CNS EC genes associated with BBB.

It is unknown how Wnt regulates BBB genes. We demonstrated that inhibiting the active Wnt pathway at E13.5 can induce the BBB gene expression pattern associated with BBB maintenance. We demonstrate that the low Wnt pathway in E13.5 causes epigenetic modifications on BBB genes mediated by HDAC2. However, the link between Wnt and epigenetic mechanisms is unclear. Since inhibiting β-catenin induces the aforementioned results, it is likely that these effects are mediated through β-catenin. In support of this model, β-catenin GOF during BBB development prevents the maturation of BBB and inhibits the differentiation of angiogenic vessels, resulting in increased brain vascularization. Similar morphological characteristics between *Hdac2* ECKO and β-catenin GOF indicate its close association.

Wnt signaling in CNS ECs is believed to require Wnt7a and Wnt7b produced by neural progenitors (*Cho et al., 2017*; *Eubelen et al., 2018*; *Kuhnert et al., 2010*; *Vallon et al., 2018*). Our transcriptomic data revealed significant expression of Wnt7a in E13.5 ECs with null expression in adults. Interestingly, adult CNS ECs showed considerable activation of Wnt7a mRNA with MS-275 treatment (*Figure 7— figure supplement 1A*). Moreover, β-catenin KD and GOF affect the expression of Wnt7a, and β-catenin staining after MS-275 treatment showed localization of β-catenin to the nucleus (*Figure 7— figure supplement 1B-D*). These data indicate a possible innate mechanism of ECs to regulate the Wnt pathway.

A natural consequence of targeting epigenetic regulators or the Wnt pathway is that these mechanisms regulate numerous genes. Our mRNA sequencing results on embryonic and adult ECs revealed that HDAC2 inhibition or deletion could regulate EC genes associated with critical functions, including angiogenesis, barrier genesis, and Wnt signaling. Our *Hdac2* ECKO phenotypic and transcriptomic data (*Figures 4A–C and 7*) reveal that the absence of HDAC2 impairs vascular and BBB maturation through the antirepression of BBB, angiogenesis, and Wnt target genes (*Figure 7A*). The resulting increase in angiogenesis and BBB permeability in *Hdac2* ECKO embryos supports our hypothesis that HDAC2-mediated epigenetic repression is essential for proper BBB and vascular development. However, our embryonic data showed no difference in *Cldn1* and *Cldn5* with *Hdac2* deletion, which differs from the adult ECs data. This discrepancy can be attributed to developmental stage, culture-induced changes in adult ECs, pan class-I HDAC inhibitor use, heterogeneous EC population, and sequencing depth.

Epidrugs are attractive therapeutics since epigenetic changes are reversible, with the potential to reestablish function after treatment. Our results support this, as the expression of *Cldn1* and *Cldn5* returns to normal after 7 d of treatment. Furthermore, our results illustrate the potential of MS-275 to reinstate the developmental characteristics in mice and partially in human adult CNS ECs.

## Methods

### Animals

Wild-type C57BL/6 J was used for embryonic and adult EC expression profiles. C57BL6/J was also used for time point analysis. *Hdac2^lox/lox^*(Strain #:022625) and *EZH2^lox/lox^*(Strain #:022616) mice were purchased from Jackson laboratories. Tamoxifen-inducible driver *Cdh5^CreERT2^* (Ralf Adams) was transferred to the PI facility from UT Southwestern. *Ctnnb1^lox/lox^* (Maketo M Taketo) was transferred from the Vanderbilt School of Medicine. Timed pregnant mice were obtained following overnight mating (the day of vaginal plug was defined as embryonic day 0.5).

### Generation of mutants

For EC-specific embryonic deletion of *Hdac2* and *Ezh2*, or β-catenin constitutive activation, male *Hdac2^lox/lox^:Cdh5^CreERT2^*, *Ezh2^lox/lox^:Cdh5^CreERT2^* and *Ctnnb1^lox/lox^:Cdh5^CreERT2^* mice were crossed to respective *^lox/lox^* females. Pregnant female mice were treated with three doses of Tamoxifen (2 mg/10 g body weight, # T5648 Millipore Sigma USA.) in corn oil through IP injection on alternate days starting E-12.5. Embryonic brains are then harvested on E-17.5 and genotyped to determine the W.T and

mutants. Embryos without Cre were selected as W.T., while *Hdac2^lox/lox:Cdh5^CreERT2*, *Ezh2^lox/lox:Cdh-5^CreERT2* and *Ctnnb1^lox/lox:Cdh5^CreERT2* embryos were identified as mutants. Animal experiments were in full compliance with the NIH guide for the care and use of laboratory animals and were approved by the UTHealth Houston Animal Welfare Committee.

## Primary brain cortical endothelial cell isolation

Embryonic or adult (male 2–3- month-old) brains were dissected under a stereo microscope and the cortex was removed. Pial membranes were peeled out of embryonic brains. For the adult brain meninges were removed by rolling the brain in filter paper. The remaining cortex without pial membrane/meninges was pooled. Isolation and culture of EC were performed according to the published methodology (Kumar T and *Kumar T and Vasudevan, 2014*; *Navone et al., 2013*). ECs were plated into type-1 collagen-coated dishes, and EC culture media from corning (#355054) was used to expand the culture. The purity of endothelial cell cultures was established with endothelial cell markers such as CD31 and isolectin B4 and determined to be more than 90%. ECs are maintained in culture for a maximum of 4 wk, and the second passage was used for the experiments.

## Drug treatments and siRNA knockdown

Adult primary cortical ECs were treated with PAN HDAC inhibitor Trichostatin A (200 nm, Selleck USA: S1045,), Class-I HDAC inhibitor MS-275 (10 µm, Selleck USA: S1053), Class-II HDAC inhibitor MC-1568(10 µm, Selleck USA:S1484), DZNeP (1 µm, Selleck USA: S7120), and Wnt3a (200 ng/ml Peprotech:315–20,) for 48 hr. E-13.5 primary ECs were treated with LF3 (50 µm, Selleck USA:S8474) for 48 hr. esiRNA specific for HDAC2, EZH2, and β-catenin were purchased from Sigma. siRNA knock-down was performed using the Lipofectamine 2000 protocol and 700 ug of siRNA was used for each transfection. Cells were washed with PBS and collected using TRIZOL for RNA extraction.

## Purification of CD31+ve ECs from E17.5 WT and *Hdac2* ECKO brain

The *Crouch and Doetsch, 2018* protocol, with modifications, was used for isolating endothelial cells from the E 17.5mouse brain cortex. Briefly, mice brains were collected into a Petri dish filled with cold 1×PBS on ice. After removing pial vessels, the cortex was isolated into HBSS/BSA/glucose solution containing DNase 1 on ice, and the individual cell suspension was obtained by trituration. Cell suspension was incubated with antibodies for 20 min on ice without agitation [CD31-APC (1:50, BD:551262), CD41-PE (1:200, BD: 558040) and CD45-PE (1:200, BD: 553081)]. Excess unbound antibodies were removed by adding extra HBSS/BSA/glucose buffer and centrifuging at 300g at 4°C for 5min. Cells were suspended in a DAPI working solution, and filtered using a 40µm cell strainer and taken for FACS sorting. During sorting, unstained controls and single-color controls were used to set the FACS parameter. Forward-scatter-A (FSC-A) and the Side-scatter-A (SSC-A) channels were used to exclude debris and gated the population that is DAPI-negative to exclude dead cells. CD45+ andCD41+ cells were excluded and CD31+ endothelial cells (CD31+ CD41−CD45−) were collected for sequencing.

## RNA-sequencing

Total RNA was isolated from E13.5, adult primary cortical ECs, and MS275-treated (10 µm) adult primary cortical ECs using Direct-zol RNA MiniPrep (Zymo research R2052). RNA samples were submitted to the Cancer Genomics Center at The University of Texas Health Science Center at Houston (CPRIT RP180734). Total RNA was quality-checked using Agilent RNA 6000 Pico kit (#5067–1513) by Agilent Bioanalyzer 2100 (Agilent Technologies, Santa Clara, USA). RNA with an integrity number greater than 7 was used for library preparation. Libraries were prepared with KAPA mRNA HyperPrep (KK8581, Roche) and KAPA UDI Adapter Kit 15 µM (KK8727, Roche) following the manufacturer's instructions. The quality of the final libraries was examined using Agilent High Sensitive DNA Kit (#5067–4626) by Agilent Bioanalyzer 2100 (Agilent Technologies, Santa Clara, USA), and the library concentrations were determined by qPCR using Collibri Library Quantification kit (#A38524500, Thermo Fisher Scientific). The libraries were pooled evenly and went for the paired-end 75-cycle sequencing on an Illumina NextSeq 550 System (Illumina, Inc, USA) using High Output Kit v2.5 (#20024907, Illumina, Inc, USA). Reads were first quality-checked by FastQC. After the QC process, the reads were aligned to the mm10 version assembly of the mouse reference genome using the STAR aligner (*Dobin et al., 2013*). The gene-level expression was quantified by counting mapped reads on each protein-coding gene

annotated by the GENCODE Project's (*Harrow et al., 2012*) M23 release for mm10 genome assembly using in-house scripts. Differentially expressed genes (DEG) were identified using DESeq2 (*Love et al., 2014*). ECs from a single litter were pooled and considered biological replicate. Three biological replicates were used for conditions E-13.5, adult, and two replicates for MS-275 groups. In DEG analysis, all samples were input to DESeq with the experimental design matrix to normalize across batch effects. The scored DEG list was filtered by selecting genes with adjusted p-values smaller than 0.05 and log fold-change greater than 0.2.

## Ultra-low mRNA sequencing

The sorted cells were frozen on dry ice as soon as the sorting is done and sent to The University of Texas Health Science Center at Houston (CPRIT RP180734). Libraries were prepared with SMART-Seq V4 PLUS Kit (R400752, Takara Bio, Japan) following the manufacturer's instructions. The quality of the final libraries was examined using Agilent High Sensitive DNA Kit (#5067–4626) by Agilent Bioanalyzer 2100 (Agilent Technologies, Santa Clara, USA) and the library concentrations were determined by qPCR using Collibri Library Quantification kit (#A38524500, Thermo Fisher Scientific). The libraries were pooled evenly and went for the paired-end 150-cycle sequencing on an Illumina NovaSeq X plus System (Illumina, Inc, USA). FastQ files were analyzed using RNA Detector (*La Ferlita et al., 2021*), with alignment performed via HISAT, followed by differential expression analysis using DESeq.

## Real-time PCR analysis

Total RNA was extracted from cells or brain samples using the Trizol reagent (Invitrogen). RNA was reverse transcribed using High capacity cDNA reverse transcription kit (Applied biosystem) for RT–qPCR according to the manufacturer's instructions. RT–qPCR was carried out on an ABI Quantdstudio 3 Real-Time PCR System using the SYBR Green method (Gendepot). RNA expression was calculated using the comparative Ct method $2-(\Delta\Delta Ct)$ normalized to *Gapdh*. Primer sequences will be available upon request. All experiments were carried out in technical duplicates and independently performed at least three times.

## Quantification of vessel percentage area and BBB leakage in the brain of embryonic pups

The method is based on the protocol published early (*Ben-Zvi et al., 2014*). To assess the BBB leakage in embryos (E-17.5), the pregnant mouse was anesthetized under isoflurane (as per the protocol). The abdominal wall was opened up to expose the embryos that were injected with lysine-fixable FITC-labeled dextran (70 KDa) into the liver using a Hamilton syringe while still attached to the mother's blood circulation. After 3–5 min of circulation, embryonic heads were dissected out and fixed by immersing in 4% PFA overnight at 4 °C. Brains were dissected out and mold was prepared using agarose and sectioned into 35 µm thick free-floating coronal slices using a semiautomatic vibratome (LEICA VT 1000 S, Leica Biosystems, Deer Park, IL, USA). Sections were co-stained with isolectin B4 to visualize blood vessels. All embryos from each litter were injected blind before genotyping. Sections were mounted using a DAPI Fluoromount-G mounting medium and imaged using a fluorescent microscope (Leica Thunder imager). Matching sections were taken from W.T and mutant brains, examined for BBB leakage, and the extravasation of FITC-dextran into the brain parenchyma was quantified using Image J software. For calculating the percentage of the area occupied by vessels in W.T and mutant brains, matching sections were selected and uploaded to AngioTool software (*Zudaire et al., 2011*) to calculate the vessel percentage area (%vessel detected/total area of the microscopic image) from each brain section stained.

## CHIP-qPCR

Quantitative chromatin immunoprecipitation (ChIP) analyses were performed and validated as described previously (*Im et al., 2004*). Primers were designed to amplify approximately 100–200 bp around the indicated region (–500, TSS, +500). The PCR primer sequences will be available upon request. The following ChIP-validated antibodies were used in the study: HDAC2 (Invitrogen: 51–5100), EED (Millipore Sigma: 17–10034), H3K27me3 (Millipore Sigma: 17–622), H3K9me3, H3K9ac (Millipore Sigma: 17–625), and H3K4me3 (Millipore Sigma: 17–614). Relative to the input DNA the enrichment

of respective epigenetic markers were quantified. To compare DNA from two different conditions, fold enrichment was calculated by further normalization of the enrichment against the IgG.

## Immunohistochemistry

For Axin2 immunohistochemistry, tissues were fixed with 4% paraformaldehyde (PFA) at 4 °C overnight. For β-catenin immunofluorescence, cells after treatment were fixed in 4% PFA for 10 min. Tissue sections or cells were blocked with 3% bovine serum albumin, permeabilized with 0.5% Triton X-100, and stained with the following primary antibodies: Axin2 (1:500, ProSci: 6163), isolectin B4 (1:200, Vector Laboratories, USA:DL-1207), β-catenin (1:500, Cell Signaling Technology USA:8480), followed by 488 Alexa Fluor-conjugated secondary antibodies (1:300– 1:1000, Invitrogen, USA:). Slides were mounted in Fluoromount G with DAPI (SouthernBiotech) and visualized under a Leica thunder fluorescence microscope.

## Human sample collection

Human cortical brain vessel samples were collected after epilepsy patients underwent temporal lobectomy. A protocol approved by the Committee for the Protection of Human Subjects was used to collect samples. Vessels are transported in an endothelial cell growth medium. The vessels were then flushed with sterile PBS to remove any blood cells and washed twice in PBS. Vessel specimens were then cut in half and seeded into the 12-well plate with a thin layer of matrigel coating. Vessels are then treated with vehicle DMSO and MS-275 for 5 d. After the treatment, vessels were washed and collected in Trizol followed by homogenization and RNA isolation.

## Statistical analysis

Statistical analysis was performed using the Sigma stat software package. All statistical tests use biological replicates and are indicated by group size (n) in figure legends. Results were expressed as mean ± SD. (standard deviation). Significance ($p < 0.05$) between two groups was calculated using an unpaired Student's t-test (two-tailed) or paired Student's t-test for two groups of values representing paired observations. One-way ANOVA assessed the statistical significance of multiple-group comparisons along with *post hoc* Bonferroni's or Tukey's multiple comparison tests. Bonferroni's Multiple Comparison Test was used for groups with different experimental numbers, while Tukey's Multiple Comparison Test was used for groups with the same experimental numbers.

## Acknowledgements

We thank Ralf Adams (Max Planck Institute for Molecular Biomedicine) and Ondine Cleaver (The University of Texas Southwestern) for providing the Cdh5(PAC)[Creert2] mice. We also thank Maketo Taketo (Kyoto University Hospital) for providing the Ctnnb1[lox/lox] mice. This work was also supported by the American Heart Association Career Development Award (18CDA34110036), NIH grant R01 to PKT (R01NS121339), and UTHealth Houston startup funds.

## Additional information

### Funding

| Funder | Grant reference number | Author |
|---|---|---|
| National Institute of Neurological Disorders and Stroke | R01NS121339 | Peeyush Kumar T |
| National Institute of Neurological Disorders and Stroke | R21NS135176 | Peeyush Kumar T |
| American Heart Association | 18CDA34110036 | Peeyush Kumar T |

| Funder | Grant reference number | Author |
|--------|------------------------|--------|

The funders had no role in study design, data collection and interpretation, or the decision to submit the work for publication.

## Author contributions

Jayanarayanan Sadanandan, Data curation, Formal analysis, Validation, Investigation, Methodology; Sithara Thomas, Formal analysis, Validation, Investigation, Methodology; Iny Elizabeth Mathew, Validation, Investigation; Zhen Huang, Conceptualization, Supervision; Spiros L Blackburn, Emery H Bresnick, Pramod K Dash, Supervision, Writing – review and editing; Nitin Tandon, Resources; Hrishikesh Lokhande, Lalit K Ahirwar, Dania Jose, Ari C Dienel, Hussein A Zeineddine, Sungha Hong, Methodology; Pierre D McCrea, Devin W McBride, Writing – review and editing; Arif Harmanci, Visualization, Methodology; Peeyush Kumar T, Conceptualization, Resources, Data curation, Formal analysis, Supervision, Funding acquisition, Validation, Investigation, Methodology, Writing – original draft

## Author ORCIDs

Nitin Tandon https://orcid.org/0000-0002-2752-2365
Emery H Bresnick https://orcid.org/0000-0002-1151-5654
Peeyush Kumar T https://orcid.org/0000-0002-5681-6602

## Ethics

This study was performed in strict accordance with the recommendations in the Guide for the Care and Use of Laboratory Animals of the National Institutes of Health. All of the animals were handled according to approved institutional animal care and use committee of the University Of Texas Health Science center at Houston. The protocol was approved by the Committee on the Ethics of Animal Experiments of the University Of Texas Health Science center at Houston. (Permit Number: AWC-22-0108). All surgery was performed under anesthesia, and every effort was made to minimize suffering.

Reviewer #1 (Public review): https://doi.org/10.7554/eLife.86978.3.sa1
Reviewer #2 (Public review): https://doi.org/10.7554/eLife.86978.3.sa2
Author response https://doi.org/10.7554/eLife.86978.3.sa3

# Additional files

## Supplementary files

- MDAR checklist
- Supplementary file 1. QPCR and Chip-qPCR primers.

## Data availability

Sequencing data have been deposited in GEO under accession codes GSE273880, GSE255967, GSE214923.

The following datasets were generated:

| Author(s) | Year | Dataset title | Dataset URL | Database and Identifier |
|-----------|------|---------------|-------------|-------------------------|
| Peeyush Kumar Thankamani Pandit | 2022 | Epigenetic Mechanisms Regulating the Transcription of BBB Genes: Role of Wnt/β-catenin signaling | https://www.ncbi.nlm.nih.gov/geo/query/acc.cgi?acc=GSE214923 | NCBI Gene Expression Omnibus, GSE214923 |

*Continued on next page*

*Continued*

| Author(s) | Year | Dataset title | Dataset URL | Database and Identifier |
|---|---|---|---|---|
| Peeyush Kumar Thankamani Pandit | 2024 | Epigenetic Mechanisms Regulating the Transcription of BBB Genes: Role of Wnt/β-catenin signaling | https://www.ncbi.nlm.nih.gov/geo/query/acc.cgi?acc=GSE255967 | NCBI Gene Expression Omnibus, GSE255967 |
| Peeyush Kumar Thankamani Pandit | 2024 | Establishing And Maintaining The Blood-Brain Barrier: Epigenetic And Signaling Determinants | https://www.ncbi.nlm.nih.gov/geo/query/acc.cgi?acc=GSE273880 | NCBI Gene Expression Omnibus, GSE273880 |

The following previously published datasets were used:

| Author(s) | Year | Dataset title | Dataset URL | Database and Identifier |
|---|---|---|---|---|
| Hupe M, Kneitz S, Davydova D, Yokota C, Kele-Olovsson J, Hot B, Gessler M, Stenman JM, MX Li | 2016 | Gene expression profiles of brain endothelial cells during embryonic development at bulk and single-cell levels | https://www.ncbi.nlm.nih.gov/geo/query/acc.cgi?acc=GSE79306 | NCBI Gene Expression Omnibus, GSE79306 |
| Hupe M, Stenman JM, MX Li | 2013 | Evaluation of TRAP-sequencing technology with a versatile conditional mouse model | https://www.ncbi.nlm.nih.gov/geo/query/acc.cgi?acc=GSE51619 | NCBI Gene Expression Omnibus, GSE51619 |

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
