## [Editor Report · eLife Assessment]

The specific questions taken up for study by the authors-in mice of HDAC and Polycomb function in the context of vascular endothelial cell (EC) gene expression relevant to the blood-brain barrier, (BBB)-are potentially **useful** in the context of vascular diversification in understanding and remedying situations where BBB function is compromised. The strength of the evidence presented is **incomplete**, and to elaborate, it is known that the culturing of endothelial cells can have a strong effect on gene expression.

---

## [Referee Report · Reviewer #1 (Public review)]

The blood-brain barrier separates neural tissue from blood-borne factors and is important for maintaining central nervous system health and function. Endothelial cells are the site of the barrier. These cells exhibit unique features relative to peripheral endothelium and a unique pattern of gene expression. There remains much to be learned about how the transcriptome of brain endothelial cells is established in development and maintained throughout life.

The manuscript by Sadanandan, Thomas et al. investigates this question by examining transcriptional and epigenetic changes in brain endothelial cells in embryonic and adult mice. Changes in transcript levels and histone marks for various BBB-relevant transcripts, including Cldn5, Mfsd2a and Zic3 were observed between E13.5 and adult mice. To perform these experiments, endothelial cells were isolated from E13.5 and adult mice, then cultured in vitro, then sequenced. This approach is problematic. It is well-established that brain endothelial cells rapidly lose their organotypic features in culture (https://elifesciences.org/articles/51276). Indeed, one of the primary genes investigated in this study, Cldn1, exhibits very low expression at the transcript level in vivo, but is strongly upregulated in cultured ECs.

(https://elifesciences.org/articles/36187 ; https://markfsabbagh.shinyapps.io/vectrdb/)

This undermines the conclusions of the study. While this manuscript is framed as investigating how epigenetic processes shape BBB formation and maintenance, they may be looking at how brain endothelial cells lose their identity in culture.

An additional concern is that for many experiments, siRNA knockdowns are performed without validation of the efficacy of knockdown.

Some experiments in the paper are promising, however. For example, the knockout of HDAC2 in endothelial cells resulting in BBB leakage was striking. Investigating the mechanisms underlying this phenotype in vivo could yield important insights.

---

## [Referee Report · Reviewer #2 (Public review)]

Sadanandan et al describe their studies in mice of HDAC and Polycomb function in the context of vascular endothelial cell (EC) gene expression relevant to the blood-brain barrier, (BBB). This topic is of interest because the BBB gene expression program represents an interesting and important vascular diversification mechanism. From an applied point of view, modifying this program could have therapeutic benefits in situations where BBB function is compromised.

The study involves comparing the transcriptomes of cultured CNS ECs at E13 and adult stages and then perturbing EC gene expression pharmacologically in cell culture (with HDAC and Polycomb inhibitors) and genetically in vivo by EC-specific conditional KO of HDAC2 and Polycomb component EZH2.

This reviewer has several critiques of the study.

First, based on published data, the effect of culturing CNS ECs is likely to have profound effects on their differentiation, especially as related to their CNS-specific phenotypes. Related to this, the authors do not state how long the cells were cultured.

Second, the use of qPCR assays for quantifying ChIP and transcript levels is inferior to ChIPseq and RNAseq. Whole genome methods, such as ChIPseq, permit a level of quality assessment that is not possible with qPCR methods. The authors should use whole genome NextGen sequencing approaches, show the alignment of reads to the genome from replicate experiments, and quantitatively analyze the technical quality of the data.

Third, the observation that pharmacologic inhibitor experiments and conditional KO experiments targeting HDAC2 and the Polycomb complex perturb EC gene expression or BBB integrity, respectively, is not particularly surprising as these proteins have broad roles in epigenetic regulation is a wide variety of cell types.

---

## [Author Response]

The following is the authors’ response to the original reviews.

Reviewers' 1 and 2 concern on endothelial cells (ECs) transcription changes on culture.

We have now addressed this concern by FACS-sorting ECs (Fig. 7A revised) and comparing our data with previous studies (S. Fig. 1C). Our major claim was the epigenetic repression of EC genes, including those involved in BBB formation and angiogenesis, during later development. To further strengthen our claim, we knocked out HDAC2 during the later stages of development to prevent this epigenetic repression. As shown in the first version of the manuscript, this knockout results in enhanced angiogenesis and a leaky BBB.

In the revised version, we have FACS-sorted CD31+ ECs from E-17.5 WT and HDAC2 ECKO mice, followed by ultra-low mRNA sequencing. Confirming the epigenetic repression via HDAC2, the HDAC2-deleted ECs showed high expression of BBB genes such as ZO-1, OCLN, MFSD2A, and GLUT1, and activation of the Wnt signaling pathway as indicated by the upregulation of Wnt target genes such as Axin2 and APCDD1. Additionally, to validate the increased angiogenesis phenotype observed, angiogenesis-related genes such as VEGFA, FLT1, and ENG were upregulated.

Since the transcriptomics of brain ECs during developmental stages has already been published in Hupe et al., 2017, we did not attempt to replicate this. However, we compared our differentially regulated genes from E-13.5 versus adult stages with the transcriptome changes during development reported by Hupe et al., 2017. We found a significant overlap in important genes such as CLDN5, LEF1, ZIC3, and MFSD2A (S. Fig. 1C).

As pointed out by the reviewer, culture-induced changes cannot be ruled out from our data. We have included a statement in the manuscript: "Even though we used similar culture conditions for both embryonic and adult cortical ECs, culture-induced changes have been reported previously and should be considered as a varying factor when interpreting our results."

Reviewer-1 Comment 2- An additional concern is that for many experiments, siRNA knockdowns are performed without validation of the efficacy of the knockdown.

We have now provided the protein expression data for HDAC2 and EZH2 in the revised manuscript Supplementary Figure- 2A.

Reviewer-1 Comment 3- Some experiments in the paper are promising, however. For example, the knockout of HDAC2 in endothelial cells resulting in BBB leakage was striking. Investigating the mechanisms underlying this phenotype in vivo could yield important insights.

We appreciate your positive comment. The in vivo HDAC2 knockout experiment serves as a validation of our in vitro findings, demonstrating that the epigenetic regulator HDAC2 can control the expression of endothelial cell (EC) genes involved in angiogenesis, blood-brain barrier (BBB) formation, and maturation. To investigate the mechanism behind the underlying phenotype of HDAC2 ECKO, we performed mRNA sequencing on HDAC2 ECKO E-17.5 ECs and discovered that vascular and BBB maturation is hindered by preventing the epigenetic repression of BBB, angiogenesis, and Wnt target genes (Fig. 7A). As a result, the HDAC2 ECKO phenotype showed increased angiogenesis and BBB leakage. This strengthens our hypothesis that HDAC2-mediated epigenetic repression is critical for BBB and vascular maturation.

Reviewer 2 Comment-2 The use of qPCR assays for quantifying ChIP and transcript levels is inferior to ChIPseq and RNAseq. Whole genome methods, such as ChIPseq, permit a level of quality assessment that is not possible with qPCR methods. The authors should use whole genome NextGen sequencing approaches, show the alignment of reads to the genome from replicate experiments, and quantitatively analyze the technical quality of the data.

We appreciate the reviewer's comment. While whole-genome methods like ChIP-seq offer comprehensive and high-throughput data, ChIP-qPCR assays remain valuable tools due to their sensitivity, specificity, and suitability for validation and targeted analysis. Our ChIP analysis identify the crucial roles of HDAC2 and PRC2, two epigenetic enzymes, in CNS endothelial cells (ECs). In vivo data presented in Figure 4 further support this finding through observed phenotypic differences. We concur that a comprehensive analysis of HDAC2 and PRC2 target genes in ECs is essential. A comprehensive analysis of HDAC2 and PRC2 target genes in ECs is currently underway and will be the subject of a separate publication due to the extensive nature of the data.

Reviewer 2 Comment-3 Third, the observation that pharmacologic inhibitor experiments and conditional KO experiments targeting HDAC2 and the Polycomb complex perturb EC gene expression or BBB integrity, respectively, is not particularly surprising as these proteins have broad roles in epigenetic regulation in a wide variety of cell types.

We appreciate the comments from the reviewers. Our results provide valuable insights into the specific epigenetic mechanisms that regulate BBB genes It is important to recognize that different cell types possess stage-specific distinct epigenetic landscapes and regulatory mechanisms. Rather than having broad roles across diverse cell types, it is more likely that HDAC2 (eventhough there are several other class and subtypes of HDACs) and the Polycomb complex exhibit specific functions within the context of EC gene expression or BBB integrity.

Moreover, the significance of our findings is enhanced by the fact that epigenetic modifications are often reversible with the assistance of epigenetic regulators. This makes them promising targets for BBB modulation. Targeting epigenetic regulators can have a widespread impact, as these mechanisms regulate numerous genes that collectively have the potential to promote the vascular repair.

A practical advantage is that FDA-approved HDAC2 inhibitors, as well as PRC2 inhibitors such as those mentioned in clinical trials NCT03211988 and NCT02601950, are already available. This facilitates the repurposing of drugs and expedites their potential for clinical translation.